# DISCOVER: Automated Curricula for Sparse-Reward Reinforcement Learning

**Leander Diaz-Bone**[*,1]    **Marco Bagatella**[*,1,2]    **Jonas Hübotter**[*,1]    **Andreas Krause**[1]

[1]ETH Zürich, Switzerland    [2]Max Planck Institute for Intelligent Systems, Germany

## Abstract

Sparse-reward reinforcement learning (RL) can model a wide range of highly complex tasks. Solving sparse-reward tasks is RL's core premise—requiring efficient exploration coupled with long-horizon credit assignment—and overcoming these challenges is key for building self-improving agents with superhuman ability. Prior work commonly explores with the objective of solving *many* sparse-reward tasks, making exploration of *individual* high-dimensional, long-horizon tasks intractable. We argue that solving such challenging tasks requires solving simpler tasks that are *relevant* to the target task, i.e., whose achieval will teach the agent skills required for solving the target task. We demonstrate that this sense of direction, necessary for effective exploration, can be extracted from existing RL algorithms, without leveraging any prior information. To this end, we propose a method for *directed sparse-reward goal-conditioned very long-horizon RL* (DISCOVER), which selects exploratory goals in the direction of the target task. We connect DISCOVER to principled exploration in bandits, formally bounding the time until the target task becomes achievable in terms of the agent's initial distance to the target, but independent of the volume of the space of all tasks. We then perform a thorough evaluation in high-dimensional environments. We find that the directed goal selection of DISCOVER solves exploration problems that are beyond the reach of prior state-of-the-art exploration methods in RL.

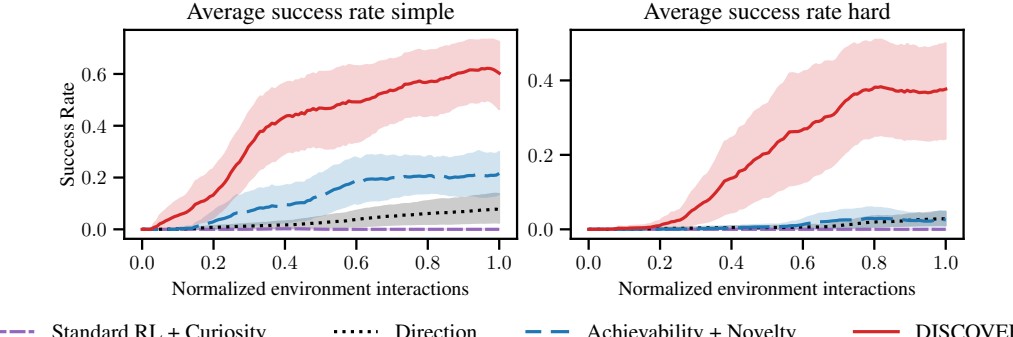

Figure 1: Given a hard task, we compare agents learning to solve this target task by learning from the experience on simpler exploratory tasks. DISCOVER uses a bootstrapped sense of direction to design a curriculum of achievable and novel exploratory tasks that are relevant to the target task. In this way, the agent bootstraps to solve much harder tasks than if using other methods for selecting exploratory tasks, such as considering only direction or only achievability and novelty. State-of-the-art standard RL algorithms using intrinsic curiosity for exploration fail to achieve our target tasks at all.

---

*Equal contribution. Correspondence to Leander Diaz-Bone <ldiazbone@ethz.ch>.

39th Conference on Neural Information Processing Systems (NeurIPS 2025).

# 1 Introduction

Reinforcement learning (RL) provides a general framework in which agents have to learn complex tasks by interacting with their environment to maximize rewards [75]. Although RL has led to breakthroughs in many domains such as Atari games [45], board games [e.g., 66], and reasoning [26], sparse-reward problems, in which a reward is only observed once the task is completed, remain very challenging [38, 80]. These sparse-reward tasks are hard to solve as the agent needs to gain a deep understanding of the environment without ever observing a reward. As such, and since they are naturally defined without any human supervision, solving sparse-reward tasks is one of the core premises of RL. We focus on the setting where the agent is given a *single* sparse-reward problem to solve. In sparse-reward RL, traditional methods from RL fail as they need to randomly observe the reward signal to make progress, which becomes exceedingly unlikely in long-horizon tasks [e.g., 49, 87].

One technique for overcoming this challenge is to frame sparse-reward tasks as multi-goal problems, which enables the agent to learn about harder goals by generalizing from already learned simpler goals [63, 5]. In this framework, a goal needs to be specified for the policy in each episode. This goal selection step plays a crucial role in guiding the exploration of the agent [53]. Choosing a goal essentially presents its own exploration-exploitation dilemma at a higher level of abstraction: should the agent try to pursue its final goal or rather aim to reach novel, but achievable intermediate goals, expanding its horizon of expertise? Previously introduced goal selection strategies have focused largely on either exploration [e.g., 53] or exploitation [5].

In this work, we argue that goal selection should naturally address this trade-off, considering both the *novelty* of commanded goals, and thus their exploration potential, as well as their *relevance* to the final objective, while considering their *achievability*. We propose DISCOVER, a method for effectively balancing *achievability*, *relevance*, and *novelty* during goal selection. We find that these quantities can be estimated by commonly used critic networks in standard deep RL algorithms. We learn an ensemble of critics to estimate the agent's uncertainty about the direction of the final task. Intuitively, DISCOVER leverages these value estimates to determine which intermediate goals are most useful towards learning to solve the final task. We find that **DISCOVER enables RL agents to solve substantially more challenging tasks than previous exploration strategies in RL**.

We further provide a formal analysis of DISCOVER, showing that it is closely linked to upper confidence bound (UCB) sampling [68], a method for balancing exploration and exploitation in sequential decision-making. We use this connection to prove bounds on the number of episodes until the final task becomes achievable. Unlike bounds in prior work, our bound for DISCOVER is independent of the volume of the space of all goals and depends linearly on the "shortest distance" between the agent's initial state and the state at which the final task is achieved. We follow this formal discussion by an extensive evaluation of DISCOVER in three complex sparse-reward environments, ranging from loco-navigation to manipulation. We observe significant improvements in sample efficiency compared to previous state-of-the-art exploration strategies in RL.

Our contributions are:

1. We propose DISCOVER, a novel goal selection strategy for solving hard tasks. By leveraging its connections to UCB, we theoretically analyze the time until the final task becomes achievable.
2. We evaluate the empirical performance of DISCOVER on various complex control tasks and show substantially improved performance compared to state-of-the-art goal selection strategies.
3. We perform a range of ablation studies, including on the utilization of prior knowledge and on the importance of balancing achievability, novelty, and relevance in goal selection.

# 2 Related Work

**Exploration in RL** Exploration has been a central challenge in reinforcement learning since its inception. Uninformed exploration strategies, such as $\epsilon$-greedy [44] or uncorrelated noise injection [24], are crucial to the success of most practical algorithms. Further improvements can be obtained by leveraging temporally extended strategies [19], or informed criteria, such as curiosity [52, 27, 61, 70, 71], diversity [21], count-based strategies [77, 10], or unsupervised environment design [82, 17]. These strategies are normally applied without any extrinsic signal [43, 61, 70, 51], or in linear combination with a specific reward signal as defined by the MDP [10, 71]; in the latter

case, balancing the two terms is often complex. This work focuses instead on a setting in which exploration is instead achieved by attempting to solve different goals than the target goal, i.e., through exploitation with respect to a different reward, as specified in the next paragraph.

**Finding solvable subgoals for exploration** Solving *many* sparse-reward problems with RL has often been formalized as goal-conditioned [22, 58] or goal-reaching [83] RL, where we jointly learn a policy for all tasks. A key contribution can be traced back to Schaul et al. [63], which introduces *universal* value function estimators, capable of estimating returns for multiple goals. To learn these functions efficiently, Andrychowicz et al. [5] proposes a goal relabeling scheme to provide signal for goals that are achieved in hindsight. Further works have explored emerging properties of goal-conditioned value functions, introducing contrastive [22, 9] or quasimetric [83, 18, 3] approaches. Interestingly, goal-conditioned approaches enable the design of nuanced exploration strategies, as the policy can be controlled through its goal-conditioning [56, 85, 53, 23, 69, 86, 35, 43, 55]. These methods strategically select a goal from the previously achieved goals. After the goal is achieved, the methods enter a pure exploration phase, in line with the successful exploration strategy proposed by Ecoffet et al. [20]. However, previous methods are generally not directed towards a particular target goal, with some exceptions relying on strong assumptions on the distance metric [57], or on additional costly components such as generative [40, 62] or discriminative [12, 11] models. In contrast, DISCOVER introduces a notion of relevance during exploration, which is fully bootstrapped from the agent's own estimates and experience, leading to deep exploration. We further discuss the connection of DISCOVER to other work such as self-play in Appendix A.

**Test-time training** We consider the setting where the agent's objective is to solve a single, challenging target task provided at "test-time", potentially starting from a pre-trained prior. This is a form of test-time training (TTT) [73], where an agent is trained specifically for the target task at test-time. TTT on (self-)supervised signals over few gradient updates has shown success in domains such as control [32], language modeling [33, 36, 74, 8], abstract reasoning [4], and video generation [15]. More recently, test-time reinforcement learning (TTRL) [88] has used RL to iteratively self-improve an agent's policy for initially unsolvable tasks, using its own experience on simpler tasks. To our knowledge, DISCOVER is the first method demonstrating effective TTRL with extensive self-supervised exploration (millions of steps), thereby enabling agents to solve highly difficult tasks.

## 3 Problem Setting

We consider the sparse-reward reinforcement learning setting and adopt the goal-conditioned formulation [75, 63], formally described by a multi-goal Markov decision process $\mathcal{M} = (\mathcal{S}, \mathcal{A}, p, \mu_0, \mathcal{G}, g^\star)$. Here $\mathcal{S}, \mathcal{A}$, and $\mathcal{G} \subseteq \mathcal{S}$ denote the set of states, actions, and goals, respectively. Additionally, $p : \mathcal{S} \times \mathcal{A} \to \Delta(\mathcal{S})$ are the transition probabilities and $\mu_0$ is the initial state distribution. In the following, we denote by $s_0$ a random initial state sampled from $\mu_0$. The target goal, denoted by $g^\star$, is the only goal we evaluate on and therefore aim to learn. The sparse goal-conditioned reward is implicitly defined as $r(s, a; g) = -\mathbb{1}\{s \notin \mathcal{S}_g\}$ [54], where $\mathcal{S}_g \subseteq \mathcal{S}$ is the subset of the states, for which the goal $g$ is achieved. We consider the general case, where this subset is defined as $\mathcal{S}_g = \{s \in \mathcal{S} \mid d(s, g) \leq \epsilon\}$. Here, $d$ defines a standard distance metric, possibly only considering some parts of the state, such as position. A goal is considered achieved during training if we have previously observed a non-negative reward for this goal. We keep track of the set of achieved goals $\mathcal{G}_{\mathrm{ach}}$, which contains all previously achieved goals (initially $\mathcal{G}_{\mathrm{ach}} = \{s_0\}$). In practice, this is implemented by maintaining a replay buffer of observed states. We consider a fixed episode length $H$, with the episode terminating once the target goal is reached. The objective of the agent is to learn a goal-conditioned policy $\pi : \mathcal{S} \times \mathcal{G} \to \Delta(\mathcal{A})$ that maximizes its value function $V^\pi(s_0, g^\star)$,

$$V^\pi(s, g) = \mathbb{E}_\pi \left[ -\sum_{t=0}^{H-1} \mathbb{1}\{s_{t'} \notin \mathcal{S}_g \forall t' \leq t\} \right]. \tag{1}$$

We denote the optimal policy by $\pi^\star : \mathcal{S} \to \Delta(\mathcal{A})$ and use $\pi_g : \mathcal{S} \to \Delta(\mathcal{A})$ to denote the policy conditioned on $g$.

### 3.1 Goal Selection for Sparse-Reward RL

The standard non-goal-conditioned online RL loop collects data according to the current policy, often with noise injection [19]. In the goal-conditioned framework, the agent can additionally control

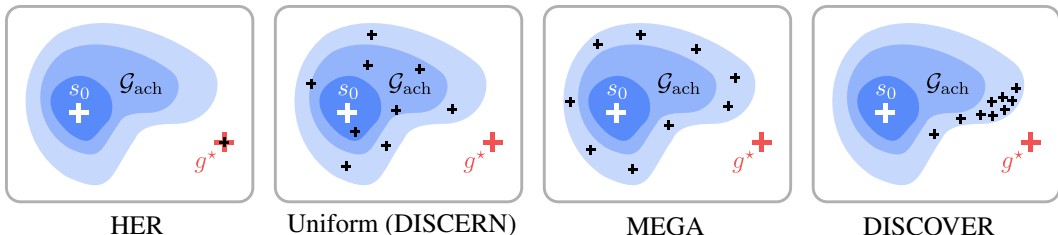

|HER|Uniform (DISCERN)|MEGA|DISCOVER|

Figure 2: Illustration of the goal selection of DISCOVER compared to prior goal selection strategies. The white cross represents the initial state of the agent, the red cross represents the target goal. The blue shaded area symbolizes the set of achieved goals $\mathcal{G}_{\text{ach}}$. A lighter blue corresponds to harder to reach goals. Finally, the black crosses represent the kinds of goals selected by each strategy.

the goal-conditioning of the policy, i.e., it may pursue arbitrary goals for data collection. We adopt the framework of Pitis et al. [53], and summarize its main steps in Algorithm 1. In each episode, SelectGoal commands a goal $g_t$ to the agent, based on prior experience, the current state of the agent and the final target goal $g^\star$. We then roll out the policy conditioned on $g_t$ until $g_t$ is achieved. After achieving the goal $g_t$, the agent enters a random exploration phase, during which it selects actions uniformly at random until the end of the episode. We add the resulting trajectory to the replay buffer. Finally, we update the agent's parameters $\theta$ using an off-policy RL algorithm, by sampling from the replay buffer and relabeling goals, as proposed by Andrychowicz et al. [5]. The focus of this work is SelectGoal, which designs the agent's curriculum. Next, we introduce DISCOVER, which selects a curriculum specifically targeted to $g^\star$.

---

**Algorithm 1** Goal-conditioned Reinforcement Learning

1: **Initialize:** replay buffer: $\mathcal{B}_0 \leftarrow \emptyset$, actor-critic parameters: $\theta_0$
2: **for** $t = 1, 2, \dots$ **do**
3:      $g_t \leftarrow \texttt{SelectGoal}(\mathcal{B}_{t-1}, \theta_{t-1}, g^\star)$            ▷ select goal $g_t$ for episode
4:      $\mathcal{B}_t \leftarrow \mathcal{B}_{t-1} \cup \{\texttt{Rollout}(\pi_{\theta_{t-1}}, g_t)\}$         ▷ add trajectory to replay buffer
5:      $\theta_t \leftarrow \texttt{Update}(\texttt{RelabelGoals}(\texttt{Sample}(\mathcal{B}_t)), \theta_{t-1})$    ▷ using off-policy RL (e.g., TD3 [24])

---

## 4 DISCOVER

We introduce DISCOVER, a method for solving hard tasks that require deep exploration. We focus on the problem of finding intermediate, solvable goals that the agent should attempt to acquire skills, eventually enabling it to achieve the target goal. We begin by presenting an intuitive explanation of the DISCOVER objective, followed by a discussion of its tight connection to the exploration-exploitation dilemma. We argue that, to efficiently learn to solve hard tasks, an agent must adhere to the following three fundamental principles:

$$\textbf{Goal utility} = \textbf{Achievability} + \textbf{Novelty} + \textbf{Relevance} \qquad (2)$$

Each of these principles is necessary to efficiently learn to solve a hard task. First, **achievability** ensures that a task is not "too hard" for the agent to ever achieve it, hence, representing meaningful experience. Second, **novelty** ensures that a task is not "too easy", such that the agent's new experience on this task is a useful learning signal to increase the policy's capabilities. Finally, **relevance** ensures that experience on a task is useful for eventually solving the target task (cf. Figure 2). Prior methods [e.g., 53] have successfully combined achievability and novelty, but are aiming to achieve all possible tasks.

After providing an intuitive discussion of the principles relevant to efficiently learning hard tasks, the question remains: how can they be quantified in practice? We propose the DISCOVER objective, which quantifies each of the discussed principles via the value estimate under the current policy. In each exploration episode, we select a goal from the achieved ones ($\mathcal{G}_{\text{ach}}$) according to

$$g_t = \underset{g \in \mathcal{G}_{\text{ach}}}{\arg\max} \; \alpha_t \Big[ \underbrace{V(s_0, g)}_{\textbf{Achievability}} + \beta_t \underbrace{\sigma(s_0, g)}_{\textbf{Novelty}} \Big] \; + \; (1 - \alpha_t) \Big[ \underbrace{V(g, g^\star) + \beta_t \, \sigma(g, g^\star)}_{\textbf{Relevance}} \Big] \qquad (3)$$

where $\alpha_t$ and $\beta_t$ are schedulable coefficients, $V$ is the mean of an ensemble of the optimal value function defined in Equation 1, and $\sigma^2$ its variance.[2] A high value $V(s_0, g)$ indicates that the policy can reach the goal $g$ from the starting state $s_0$, therefore promoting **achievability**. In contrast, a large uncertainty $\sigma(s_0, g)$ indicates that the agent is not reliably reaching the goal $g$. Hence, attempting such goals prioritizes **novel** experiences. Finally, a high value $V(g, g^\star)$ indicates that $g$ is "closely related" to the target goal $g^\star$. The **relevance** term of DISCOVER prioritizes goals that *might* be closely related to the target goal, either since they already have a high value $V(g, g^\star)$ or because the agent is still uncertain about this value. The relevance term can also be viewed as directing the goal selection towards the final target $g^\star$. Note that while we focus on a single target $g^\star$, DISCOVER is easily extended to distributions over targets by maximizing Equation (3) in expectation.

We emphasize that all terms in Equation (3) are estimated by the critic and do not rely on any prior information. In particular, the relevance estimates are entirely bootstrapped from the agent's current knowledge. While the above gives an intuitive introduction, DISCOVER can be interpreted as an agent seeking to maximize the likelihood of reaching the target goal $g^\star$, as we describe in Appendix B.

## 4.1 Automatic Online Parameter Adaptation

Prior work has explored adapting the parameters online to meet predefined metrics, such as the entropy of the policy [28, 71]. In the same spirit, we propose a simple online parameter adaptation strategy that adjusts $\alpha_t$ and $\beta_t$ to maintain a fixed target goal achievement rate $p^\star$, similarly to the cutoff strategy proposed in Pitis et al. [53]. This approach aligns with the previously discussed intuition, as it ensures that the agent neither selects goals that are "too easy" nor ones that are "too hard". We find that it is sufficient to linearly increase $\alpha_t$ when the agent achieved too few of its previously selected goals, and linearly decrease $\alpha_t$, when the agent achieved too many, while setting $\beta_t = 1$:

$$\alpha_{t+1} = \Pi_{[0,1]}(\alpha_t + \eta(p_t - p^\star)). \tag{4}$$

Here, $p_t$ denotes the average goal achievement rate over the last $k_{\text{adapt}}$ episodes, $\Pi_{[a,b]}$ clips the output value to the interval $[a, b]$, and $\eta > 0$ is the adaptation step size, which we set to $0.01$. We ablate the effects of our adaptation strategy in Figure 10 in the appendix, and find that the optimal target goal achievement rate $p^\star$ is approximately $50\%$, consistent with findings in prior work [53].

## 4.2 Connection to the Exploration-Exploitation Dilemma

Balancing exploration and exploitation is key when proposing tasks such that the resulting experience is valuable toward solving the target task. DISCOVER balances exploration and exploitation in two ways. First, in its estimate of $V(s_0, g)$, which leads to a trade-off between selecting achievable and novel goals. Second, in its estimate of $V(g, g^\star)$, where exploitation leads to directing goal selection toward the target goal while exploration prevents the agent from overfitting to an overconfident estimate of direction.

**Theoretical guarantee for DISCOVER** To illustrate the underlying exploration-exploitation trade-off, we consider a simplified setting and use the link between DISCOVER and UCB to prove rates for the number of episodes until the DISCOVER agent can achieve the target goal. We make the following simplifying regularity assumptions, analogously to the linear bandit setting [see, e.g., 2, 13]:

**Informal Assumption 4.1** (formalized in Assumptions C.1 to C.5)**.**

1. *Optimal value functions are linear in latent features:* Among the set of achievable[3] goals $g$, the value functions $V^\star(s_0, g)$ and $V^\star(g, g^\star)$ are linear in a known $d$-dimensional feature space. Note that the above only needs to hold for currently achievable goals.

2. *Noisy feedback:* In episode $t$, selecting any achievable $g_t$, the agent receives noisy feedback of the optimal value functions, with the noise being conditionally sub-Gaussian.

3. *Goal space contains optimal paths:* For any goal $g$, the optimal path from $g$ to the target goal $g^\star$ is contained in the goal space $\mathcal{G}$.[4]

4. *Goal achievability:* Based on our previous observation that $\alpha$ controls the rate of goal-achieval, we let the probability of achieving any *achievable* goal $g$ be at least $\alpha$. Any goal $g$ is considered

---

[2]See Appendix E.1.
[3]The set of *achievable* goals is defined in Assumption 4.
[4]That is, $\mathcal{G}$ is $g^\star$-geodesically convex under the forward quasimetric induced by the optimal value function.

*achievable* in episode $t$ if the agent previously achieved a goal $g_{t'}$ ($t' < t$) which is within distance $(1 - \alpha)\kappa$ of $g$ (under the optimal value function). Here, $\kappa > 0$ is a rate of expansion.

Assumptions 1 and 2 are standard in the literature on linear bandits. Note, however, that in our setting this feedback model implicitly assumes feedback on $V^\star(g_t, g^\star)$, which can be thought of as a generalization condition: If the agent cannot generalize from its experience from $s_0$ to $g$ about the relation of $g$ and $g^\star$, then it may never reach the target goal $g^\star$. Another implicit consequence of the linearity assumption is that the bias and non-stationarity of the value function estimates is controlled, which is not commonly the case when learning value functions with neural networks via bootstrapping. Assumption 3 is a continuity assumption on the goal space, which is satisfied in many practical settings. Finally, Assumption 4 leads to a trade-off in choosing $\alpha$: A small $\alpha$ leads to a larger set of achievable goals, but at a lower goal-achieval rate. In contrast, a large $\alpha$ leads to a more conservative set of achievable goals, but at a higher goal-achieval rate. This trade-off matches the empirical effect of $\alpha$ (cf. Figure 10 in the appendix). The parameter $\kappa$ controls the rate of expansion of the set of achievable goals.

In the above setting, we bound the number of episodes until DISCOVER reaches the target goal $g^\star$:

**Informal Theorem 4.2** (formalized in Theorem C.9). *Fix any confidence $\delta \in (0, 1)$ and any $\alpha \in (0, \frac{1}{2})$. Let the above assumptions hold. We denote by $D = -V^\star(s_0, g^\star)$ the distance from the initial state $s_0$ to the target goal $g^\star$ under the optimal policy. Then, with probability $1 - \delta$, selecting goals $g_t$ with DISCOVER (with $\alpha_t = \alpha$ and $\beta_t$ chosen appropriately), the number of episodes $N$ until the target goal $g^\star$ is achievable by the agent is bounded by*

$$N \leq \widetilde{O}\left(\frac{Dd^2}{\alpha(1 - 2\alpha)^2(1 - \alpha)^3 \kappa^3}\right) = \widetilde{O}\left(\frac{Dd^2}{\kappa^3}\right).$$

This theorem shows that DISCOVER efficiently learns to solve hard, initially unachievable tasks. With a larger dimensionality $d$ of the feature space, the learning task becomes harder, which is reflected in the bound. Similarly, the larger the distance $D$ from the initial state to the target goal, the longer it takes to reach the target goal. In contrast, the larger the rate of expansion $\kappa$, the faster the agent reaches the target goal, since the set of achievable goals expands faster. Finally, the bound suggests choosing $\alpha \approx 0.1$ (cf. Figure 7 in the appendix) which roughly matches the average $\alpha$ chosen in experiments by our adaptation strategy (cf. Figure 11 in the appendix).

Despite our simplifying assumptions, achieving the target goal $g^\star$ remains non-trivial. The agent must trade off learning the value function against exploiting the direction this value suggests toward $g^\star$. Overconfidence in the value estimate may lead the agent to diverge and never reach $g^\star$. On the other hand, if the agent is too conservative in its value estimate (or does not use the value estimate for direction at all), it may never reach the target goal within a reasonable time. To see why guidance matters, imagine an $m$-dimensional goal space that is a ball of radius $R$. Covering that ball with hypercubes of side $\epsilon$ requires on the order of $(R/\epsilon)^m$ cells, exponential in $m$. This back-of-the-envelope calculation indicates that to achieve hard goals in a high-dimensional goal space, undirected exploration is insufficient, and the agent must balance exploration and exploitation. In doing so, DISCOVER avoids this curse of dimensionality and exploits the learned value to stay close to a nearly one-dimensional corridor from $s_0$ to $g^\star$. For this reason, the bound in Informal Theorem 4.2 depends *solely* on the (1-dim) distance $D$ and *not* on the total volume of the goal space $\mathcal{G}$. Bounds of previous work [78] depend on the total volume of $\mathcal{G}$, and quickly become vacuous in high-dimensional problems.

## 5  Results

We evaluate the empirical performance of DISCOVER across three complex, sparse-reward, long-horizon control tasks, highlighting five main insights. Unless mentioned otherwise, we employ the TD3 [24] actor-critic algorithm for training all agents. While our evaluation focuses on model-free methods, the goal-based directed exploration of DISCOVER can also be used with model-based backbones such as Dreamer [29, 30, 31]. For all experiments, we report the mean performance across 10 seeds along with its standard error. Additional implementation details, hyperparameter choices and experimental results are reported in Appendices D, E and E.3, respectively. The code is available at https://github.com/LeanderDiazBone/discover.

**Environments**  For our experiments, we use the JaxGCRL library [9] to assess performance on challenging, high-dimensional navigation and manipulation tasks. Specifically, we evaluate on

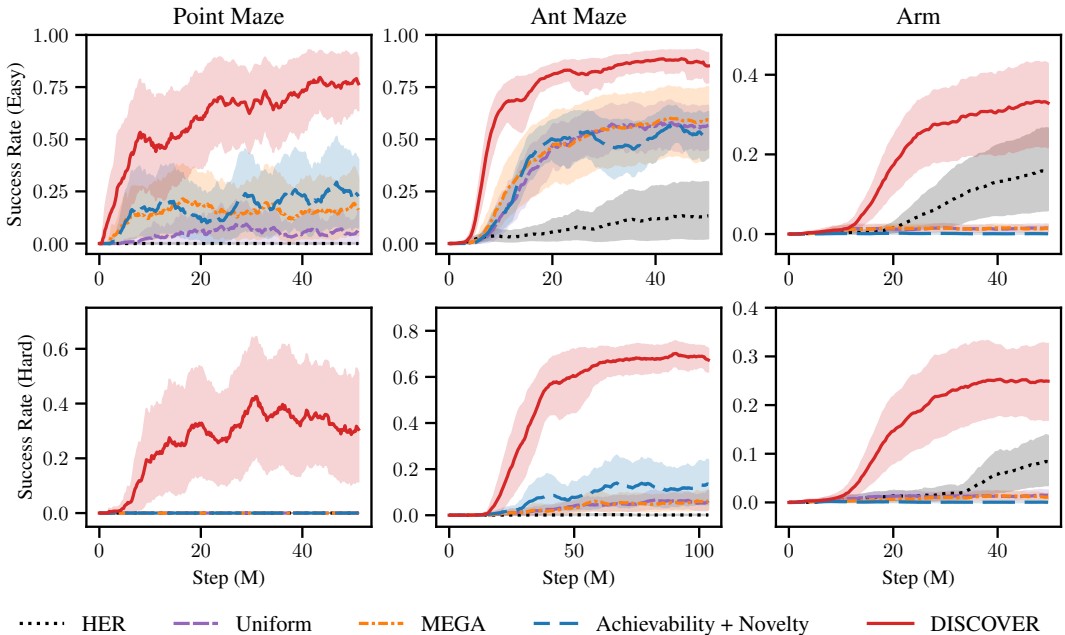

Figure 3: Comparison of the success rates on the target task over the course of training in the `pointmaze`, `antmaze` & `arm` environments. We compare DISCOVER to other strategies for goal selection. We consider two difficulty levels for each environment. We find that the DISCOVER agents learn to solve difficult target tasks significantly faster than the baselines.

the `antmaze` environment, where a simulated quadruped with a 27-dimensional state space and an 8-dimensional action space must learn to navigate through a maze to reach a target location. For manipulation, we consider the `arm` environment, which features a 23-dimensional state and a 5-dimensional action space. In this task, a robotic arm must move a cube to a specified target location, potentially while avoiding obstacles. Additionally, we evaluate on randomly generated `pointmazes` of varying dimensionality, to assess the capabilities of different goal selection strategies to explore high-dimensional goal spaces efficiently. We construct these $n$-dimensional `pointmazes` by randomly generating paths in the $n$-dimensional hypercube until the target location was reached sufficiently often from the starting location. For all environments, we consider both a simple and a hard configuration, where the latter is characterized by longer horizons, more complex obstacles, or higher-dimensional goal spaces.

**Goal selection baselines** We compare DISCOVER with the following baselines.

1. Hindsight Experience Replay (HER): The HER goal selection strategy [5] always selects goals from the target goal distribution. Since we consider a single fixed target goal, the goal at time $t$ is set as $g_t = g^\star$.

2. DISCERN (uniform): The uniform goal selection strategy [85] samples goals uniformly from the support of the achieved goal distribution, i.e., $g_t \sim \text{Unif}(\text{supp}(p_{\text{ach}}))$. The achieved goal distribution $p_{\text{ach}}$ is modeled using a kernel density estimator based on randomly sampled previously achieved goals.

3. Maximum Entropy Gain Achievement (MEGA): The MEGA goal selection strategy [53] selects goals with the lowest likelihood from the set of achievable goals, i.e., $g_t = \arg\min_{g \in \mathcal{G}_{\text{ach}}} p_{\text{ach}}(g)$. In contrast to DISCOVER, MEGA additionally defines a goal $g$ as achievable if its value function $V(s_0, g)$ exceeds a threshold, which is adapted dynamically based on the goal achievement rate over recent episodes.

4. Achievability + Novelty: This baseline corresponds to the undirected components of DISCOVER, i.e., $g_t = \arg\max_{g \in \mathcal{G}_{\text{ach}}} V(s_0, g) + \beta_t \sigma(s_0, g)$.

**Insight 1: DISCOVER outperforms state-of-the-art goal selection strategies in complex control environments.** As shown in Figure 3, DISCOVER consistently outperforms all baseline goal selection strategies across the evaluated environments. The performance gains are particularly

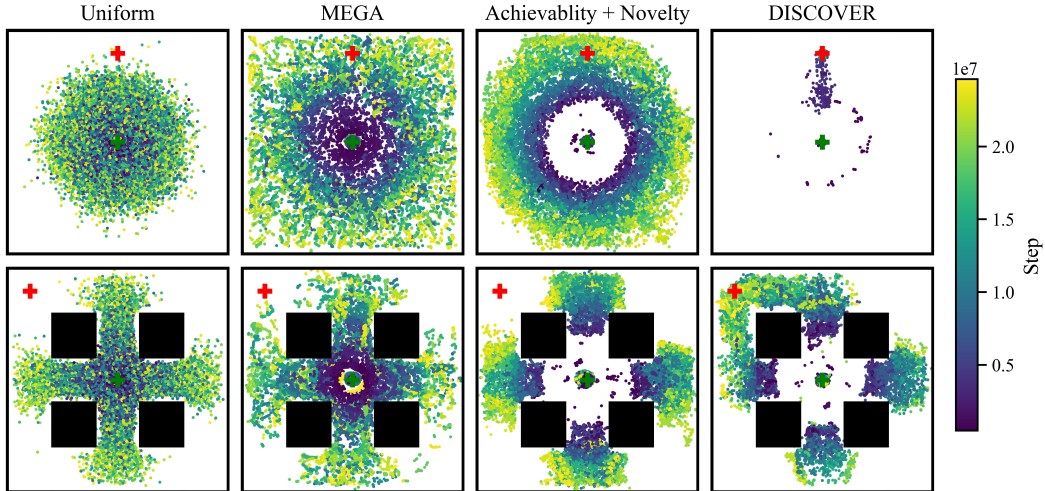

Figure 4: Visualization of the selected goals of different goal selection strategies during the first 25M steps on the `antmaze` environment, colored by time step. DISCOVER balances exploring the environment with exploiting the agent's sense of direction to select goals relevant to the target task.

notable in the more challenging variants of each task, where emphasizing *relevance* proves critical for learning optimal policies. Interestingly, in the `arm` environment, HER performs better than some undirected goal selection strategies. We attribute this to the unconstrained nature of the goal space in the `arm` environment, which leads undirected methods to explore irrelevant regions that do not contribute to learning about the target goal.

**Insight 2: Undirected goal selection is insufficient for high-dimensional search spaces.** We next evaluate the performance of goal selection strategies on `pointmaze` navigation tasks of varying dimensionality. Table 1 demonstrates that the undirected goal selection strategies are eventually successful in two dimensions, but consistently fail in dimension larger than three. In contrast, DISCOVER, by focusing on the most relevant directions, successfully solves mazes in up to six dimensions. The substantial difference in empirical performance can be explained by the size of the search space explored by each method. For undirected methods, this space grows exponentially with dimension, while DISCOVER mitigates this challenge by only selecting goals that likely to lead to the final objective, thereby dramatically reducing the effective search space. While the performance gap

| Dimension | 2 | 3 | 4 | 5 | 6 |
|---|---|---|---|---|---|
| HER | $\infty$ | $\infty$ | $\infty$ | $\infty$ | $\infty$ |
| MEGA | 4.8 | $\infty$ | $\infty$ | $\infty$ | $\infty$ |
| Ach. + Nov. | 5.2 | $\infty$ | $\infty$ | $\infty$ | $\infty$ |
| DISCOVER | **2.9** | **3.1** | **7.4** | **5.4** | **18.7** |

Table 1: Comparison of the required number of steps (M) for reaching $10\%$ target goal achievement in `pointmazes` of varying dimension. $\infty$ denotes no achievement of $10\%$ success rate before termination after 50M steps. DISCOVER scales to large mazes due to its awareness of direction and uncertainty.

is most pronounced in complex, high-dimensional tasks, we demonstrate that DISCOVER also improves performance in standard sparse-reward environments.

**Insight 3: DISCOVER improves performance by selecting relevant goals towards reaching the target, while exploring sufficiently.** We visualize the goal selection behavior of DISCOVER and the baseline strategies in the `antmaze` environment in Figure 4. While all baseline methods perform undirected goal selection, exploring the entire state space, DISCOVER, after an initial exploration phase, quickly identifies the correct direction and subsequently focuses its goal selection on the relevant region. These plots explain the improved performance, as DISCOVER can select (and achieve) goals at the target much earlier than the other strategies. We provide a detailed evaluation of how the different components of DISCOVER influence goal selection in Appendix D (cf. Figure 14).

**Insight 4: DISCOVER can leverage prior knowledge to further accelerate exploration.** In many scenarios, prior information about the environment is available, for example through pre-training on a related environment. To evaluate whether DISCOVER can leverage prior knowledge for exploration, we integrate prior information on *relevance* by substituting the prior for $V(g, g^\star)$ in Equation (3). We evaluate two kinds of priors: (1) a hand-designed prior, leveraging human knowledge akin to reward shaping, and (2) a pre-trained prior from a similar environment. We use the `antmaze` (hard) environment, picking as a natural hand-designed prior the $L_2$-distance to the target goal $g^\star$ (ignoring obstacles), and as the pre-trained prior a value function learned on a `pointmaze` with the same layout. We find in Figure 5 that DISCOVER with a prior can explore marginally faster than bootstrapping value estimates from scratch. In particular, in the maze with obstacles, using the

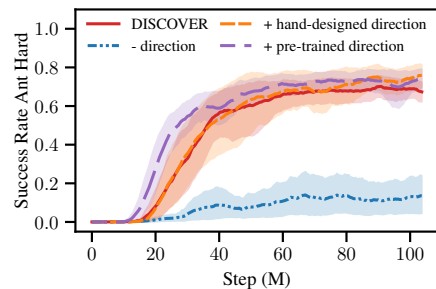

Figure 5: Comparison of using different strategies to determine direction, which replace the $V(g, g^\star)$ term in the DIS-COVER objective. Hand-designed direction: $\|g - g^\star\|_2$; pre-trained direction: critic from training in a `pointmaze` environment with the same maze layout.

pre-trained prior accelerates exploration, since the agent explores only in the direction of the target (cf. Figure 13 in the appendix). We expect that this benefit of priors increases in even harder tasks, since DISCOVER explores only the difficult and novel aspects of the target task.

**Insight 5: Subgoal selection enables deep exploration.** Finally, we evaluate the importance of goal-conditioning for solving sparse-reward tasks. To this end, we compare DISCOVER to methods for exploration in RL that do not explicitly select exploratory intermediate goals [24, 27, 48]. This includes state-of-the-art approaches based on curiosity and reward shaping [10, 71]. We show in Figure 6 that even in the simpler environments, these standard methods are unable to solve the task. In Figure 15 of the appendix, we visualize the states visited by the non-goal-conditioned baselines. While methods leveraging curiosity to shape the reward explore faster than others, none of them reaches the target task within our maximum number of episodes. This highlights that in long-horizon, sparse-reward settings, directed goal selection can facilitate deep exploration, which is essential for solving complex tasks.

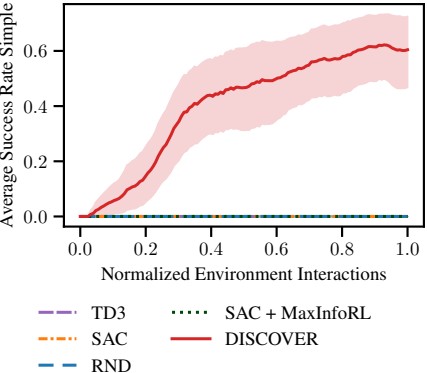

Figure 6: State-of-the-art RL algorithms fail to achieve the target tasks even in their "simple" configuration.

## 6 Conclusion

In this work, we introduce DISCOVER, a goal selection method for solving challenging tasks by balancing *novelty*, *achievability*, and *relevance*. DISCOVER is closely related to principled methods for balancing exploration and exploitation, and we theoretically show that it efficiently reaches the target goal in a simplified linear bandit setting. We further empirically evaluate DISCOVER on various complex control tasks and find that it consistently outperforms prior state-of-the-art exploration strategies in RL in solving difficult, sparse-reward tasks.

A limitation of DISCOVER is that it relies on bootstrapped estimates of the value function, which depend on the ability of the value network to generalize. Additionally, our method incurs overhead by training an ensemble of critics for uncertainty estimation, potentially limiting its direct applicability to implementations involving large critic networks such as in language domains. We believe that exploring alternative means of efficient uncertainty estimation for DISCOVER is an exciting direction for future work. Furthermore, we hypothesize that directed exploration is especially advantageous in high-dimensional and complex goal spaces, as demonstrated for the `pointmaze` with variable dimensions. Therefore, particularly interesting is the application of DISCOVER to problems with highly complex goal spaces, such as in mathematics or programming with large language model priors. Finally, our work also highlights important directions for future research, including enabling the generation (rather than selection) of goals, extending DISCOVER to hierarchical planning, and reusing experience from one target task for future tasks.

## Acknowledgments

We would like to thank Bhavya Sukhija, Yarden As, and Nico Gürtler for feedback on early versions of the paper. Marco Bagatella is supported by the Max Planck ETH Center for Learning Systems. This project was supported in part by the European Research Council (ERC) under the European Union's Horizon 2020 research and Innovation Program Grant agreement no. 815943, and the Swiss National Science Foundation under NCCR Automation, grant agreement 51NF40 180545.

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

# Appendices

# A Additional Related Work

**Self-play**  Many games can be naturally formulated as sparse-reward reinforcement learning problems, where the agent receives a reward only upon winning. Seminal works have achieved superhuman ability using this approach [e.g., 45, 66, 67, 7]. A central technique, enabling these successes, is self-play [79, 64, 65, 66]. There are clear connections between these techniques and the directed exploration in DISCOVER. Self-play can be viewed as a form of goal selection [64, 65], where the current agent (or a simultaneously trained agent) is choosen as its own opponent. This opponent selection promotes *achievability*, as it is possible to beat a recent version of yourself, *novelty* as the opponent is of similar ability, and *relevance* as it is the currently strongest opponent to beat.

**Hierarchical RL**  DISCOVER is closely related to the field of hierarchical RL [76, 16, 46], which aims to explore and plan at a higher level of abstraction. Recent methods have introduced frameworks in which multiple hierarchical levels are used to propose and learn skills [46, 25]. In this work, we focus on a simplified setting in which a single goal is selected for exploration in each episode. However, the general approach can be naturally extended to more complex hierarchical structures.

**Directed exploration in sequential decision making**  DISCOVER is designed to efficiently address the exploration-exploitation dilemma, a key concept in sequential decision making. Fundamentally, it expresses an agent's inevitable trade-off between the objectives of learning its environment and solving a task. A particularly common approach to balancing exploration and exploitation is grounded in the principle of optimism in the face of uncertainty [e.g., 68]. Thereby, the agent selects actions that maximize an upper confidence bound (UCB) of the reward function, i.e., it selects actions which based on the agents' imperfect knowledge *could* lead to a large reward. In many settings of sequential decision making, such as linear bandits, this approach achieves the rate-optimal regret $R_T \sim \tilde{O}(d\sqrt{T})$ [68, 2, 1, 13]. The DISCOVER objective extends UCB to the problem of goal selection in RL. Beyond UCB, many other methods have been shown to effectively direct exploration towards "relevant" experience, such as in bandits [e.g., 60, 59, 34], in RL [e.g., 14, 72], or in active learning [e.g., 42, 37, 6].

# B Connection to the Likelihood of Success

From a goal-reaching perspective, the undiscounted version of the DISCOVER criterion is tightly connected to the actual objective of the agent, i.e., reaching the target goal. This emerges naturally as when $\gamma \to 1$, $\epsilon \to 0$, and $\pi \to \pi^\star$, the value function becomes a (negative) quasimetric [83]. Thus, it is non-positive, and respects the triangle inequality,

$$V^\pi(s_0, g) + V^\pi(g, g^\star) \leq V^\pi(s_0, g^\star), \tag{5}$$

for arbitrary $s_0 \in \mathcal{S}, g, g^\star \in \mathcal{G}$. Note that the direction of the inequality is flipped as the value function represents a negative distance. For $\alpha = 0.5$, DISCOVER maximizes a probabilistic estimate of the left hand side of Equation (5), which is a tight lower-bound to $V^\pi(s_0, g^\star)$. The value $V^\pi(s_0, g^\star)$ is exactly the quantity of interest for the agent, as it represents the negative expected number of steps to reach the true goal $g^\star$. Intuitively, DISCOVER selects goals that are optimistically going to guarantee the shortest path to the actual goal or, in the undiscounted case, the likelihood of reaching it within an episode.

# C Proofs

In this section, we prove the theoretical guarantee informally stated in Informal Theorem 4.2. We begin by introducing some useful notation and the formal assumptions before proving Theorem C.9.

## C.1 Notation

We define the reward function $r^\star(g) = \alpha V^\star(s_0, g) + (1 - \alpha)V^\star(g, g^\star)$ for any fixed $\alpha \in (0, \frac{1}{2})$. We define $d^\star(g, g') = -V^\star(g, g^\star)$. We denote by $\mathcal{G}_t \subseteq \mathcal{G}$ the goals that are achievable by the agent in episode $t$ with probability at least $\alpha$, i.e., the policy is able to reach goals within $\mathcal{G}_t$ with probability at least $\alpha$. We use $\log$ to denote the natural logarithm. For simplicity, we assume throughout that the initial state $s_0$ is fixed across all episodes.

## C.2 Assumptions

**Assumption C.1** (Linear value function within feature space). For any $n \geq 1$ and for all $g \in \mathcal{G}_n$, the value functions $V^\star(s_0, g)$ and $V^\star(g, g^\star)$ are linear in the features $\phi(\cdot), \varphi(\cdot) \in \mathbb{R}^d$ with $\phi(\cdot) \perp \varphi(\cdot)$, i.e.,

$$V^\star(s_0, g) = \langle \phi(g), \theta \rangle \quad \text{and} \quad V^\star(g, g^\star) = \langle \varphi(g), \theta' \rangle$$

for some fixed $\theta, \theta' \in \mathbb{R}^d$ with $\|\theta\|_2, \|\theta'\|_2 \leq 1$.

**Assumption C.2** (Noisy feedback). In episode $t$, selecting any $g_t \in \mathcal{G}_t$, the agent receives noisy feedback $y_t = r^\star(g_t) + \varepsilon_t$. We assume that the noise sequence $\{\varepsilon_t\}_{t=1}^\infty$ is conditionally $R$-sub-Gaussian for a fixed constant $R \geq 0$, i.e.,

$$\forall t \geq 0, \quad \forall \lambda \in \mathbb{R}, \quad \mathbb{E}[e^{\lambda \varepsilon_t} \mid \mathcal{F}_{t-1}] \leq \exp\left(\frac{\lambda^2 R^2}{2}\right)$$

where $\mathcal{F}_{t-1}$ is the $\sigma$-algebra generated by the random variables $\{g_s, \varepsilon_s\}_{s=1}^{t-1}$ and $g_t$.

**Assumption C.3** (Value function estimates). For any $h \geq 1, t \geq h$, the value function estimate relative to $s_0$ is given by

$$V_t(s_0, \cdot) \stackrel{\text{def}}{=} \phi(\cdot)^\top (\Sigma_t + \lambda I)^{-1} \sum_{s=h}^{t-1} \phi(g_s) y_s,$$

$$\sigma_t(s_0, \cdot) \stackrel{\text{def}}{=} \sqrt{\phi(\cdot)^\top (\Sigma_t + \lambda I)^{-1} \phi(\cdot)},$$

where $\Sigma_t = \sum_{s=h}^{t-1} \phi(g_s) \phi(g_s)^\top$ and $\lambda > 0$. The value function estimate relative to $g^\star$ is defined analogously with respect to the feature vector $\varphi(\cdot)$.

**Assumption C.4** (Goal space contains optimal paths[5]). For any goal $g' \in \mathcal{G}$, the optimal path from $g'$ to $g^\star$ is contained in $\mathcal{G}$. Formally, there exists a $g \in \mathcal{G}$ such that $d^\star(g', g^\star) = d^\star(g', g) + d^\star(g, g^\star)$ for any $d^\star(g', g) \in [0, d^\star(g', g^\star)]$.

**Assumption C.5** (Goal achievability). We denote by $\mathcal{G}_t \subseteq \mathcal{G}$ the goals that are achievable by the agent in episode $t$ with probability at least $\alpha \in (0, 1)$. Moreover, for any $t \geq 1$, the $\mathcal{G}_t$ contains all goals $g \in \mathcal{G}$ for which we have previously selected a goal, which is $(1 - \alpha)\kappa$-close (under the optimal value function). We call $\kappa > 0$ the *expansion rate*. Formally,

$$\mathcal{G}_{t+1} \supseteq \{g \in \mathcal{G} \mid \exists t' \leq t : d^\star(g_{t'}, g) \leq (1 - \alpha)\kappa\}.$$

Further, $\mathcal{G}_0 \supseteq \{s_0\}$, i.e., the initial state is always achievable.

**Relaxing Assumption C.2** One can consider any individual feedback $y_t$ as being the result of an oracle that achieves the commanded goal $g_t$ with probability at least $\alpha$ within $K$ episodes. With this looser assumption, our bound in Informal Theorem 4.2 simply increases by a factor $K$.

## C.3 Proof of Informal Theorem 4.2

We begin by restating the regret bound obtained for linear bandits in [13].

**Proposition C.6.** *Let Assumptions C.1 to C.3 hold. Fix any $\delta \in (0, 1), n \geq 1, \alpha \in (0, 1)$, and let*

$$\beta_t = 1 + R\sqrt{2(d \log(t - n + 1) + 1 + \log(1/\delta))} \quad \text{for } t \geq n. \tag{6}$$

*We then have with probability $1 - \delta$ that the regret of selecting goals $g_n, g_{n+1}, \ldots$ with DISCOVER($\alpha, \beta_t$) is bounded by*

$$\sum_{t=h}^{h+T} \max_{g \in \mathcal{G}_n} r^\star(g) - r^\star(g_t) \leq O(d\sqrt{T} \log T \log(1/\delta)).$$

---

[5]This assumption simply states that the goal space $\mathcal{G}$ is geodesically convex under the quasimetric $d^\star$ induced by the optimal value function. This is a standard "reachability" condition, which may be familiar to readers from control theory.

*Proof.* By Theorem 3 in [13] and using that the feature spaces $\phi$ and $\varphi$ are orthogonal (cf. Assumption C.1), we have

$$\sum_{t=h}^{h+T} \max_{g \in \mathcal{G}_n} r^\star(g) - r^\star(g_t) \leq O(\sqrt{T}(B\sqrt{\gamma_T} + \gamma_T + \sqrt{\gamma_T} \log(1/\delta)))$$

with $B = 1$. Bounding $\gamma_T \leq O(d \log T)$ using Assumption C.1, completes the proof. $\qquad\square$

**Lemma C.7.** *Let Assumption C.4 hold and fix any $\epsilon > 0, \alpha > 0$. Then, for all $g' \in \mathcal{G}$ with $d^\star(g', g^\star) \geq \epsilon$ there exists a $g \in \mathcal{G}$ with $d^\star(g, g') = \epsilon$ such that $r^\star(g) - r^\star(g') \geq (1 - 2\alpha)\epsilon$.*

*Proof.* Consider the optimal path from $g'$ to $g^\star$, i.e., the goals $g$ satisfying

$$d^\star(g', g^\star) = d^\star(g', g) + d^\star(g, g^\star).$$

By Assumption C.4, for any $d^\star(g', g) \in [0, \epsilon]$, we have that $g \in \mathcal{G}$. We take the goal $g$ such that $d^\star(g', g) = \epsilon$. We then obtain

$$\begin{aligned}
r^\star(g) - r^\star(g') &= \alpha(V^\star(s_0, g) - V^\star(s_0, g')) + (1 - \alpha)(V^\star(g, g^\star) - V^\star(g', g^\star)) \\
&= \alpha(d^\star(s_0, g') - d^\star(s_0, g)) + (1 - \alpha)(d^\star(g', g^\star) - d^\star(g, g^\star)) \\
&= \alpha(d^\star(s_0, g') - d^\star(s_0, g)) + (1 - \alpha)\epsilon \\
&\geq \alpha(d^\star(s_0, g') - (d^\star(s_0, g') + d^\star(g', g))) + (1 - \alpha)\epsilon \quad \text{(triangle inequality)} \\
&= -\alpha d^\star(g', g) + (1 - \alpha)\epsilon \\
&= -\alpha\epsilon + (1 - \alpha)\epsilon \\
&= (1 - 2\alpha)\epsilon.
\end{aligned}$$

$\square$

**Lemma C.8** (Improvement lemma). *Let Assumptions C.1 to C.4 hold with $\beta_t$ as in Equation (6). Fix any $\delta \in (0, 1)$, $n \geq 1$, $\epsilon > 0$, $\alpha \in (0, \frac{1}{2})$ and $0 < \Delta < (1 - 2\alpha)\epsilon$. With probability $1 - \delta$, there exist a $t' \in \{n, \ldots, n + T\}$ with $T = \widetilde{\Theta}(\frac{d^2}{\alpha((1-2\alpha)\epsilon - \Delta)^2})$ and a $\tilde{g} \in \mathcal{G}$ with $d^\star(g_{t'}, \tilde{g}) \leq \epsilon$ such that*

$$r^\star(\tilde{g}) - \max_{g \in \mathcal{G}_n} r^\star(g) \geq \Delta.$$

*Proof.* With probability $1 - \frac{\delta}{2}$, the number of episodes $T$ until the agent has achieved $T_{\text{ach}}$ of its goals is bounded as $T = \widetilde{\Theta}(\frac{T_{\text{ach}}}{\alpha})$.[6] We denote by $\mathcal{T} \subseteq \{n, \ldots, n + T\}$ the set of episodes in which the agent has achieved its goal. Further, by Proposition C.6, also with probability $1 - \frac{\delta}{2}$, we have

$$R_n \overset{\text{def}}{=} \frac{1}{|\mathcal{T}|} \sum_{t \in \mathcal{T}} \max_{g \in \mathcal{G}_n} r^\star(g) - r^\star(g_t) \leq \widetilde{O}(d/\sqrt{|\mathcal{T}|}).$$

All further steps are conditional on the union of the above high probability events. Thus, the regret in successful episodes is bounded by $R_n \leq \widetilde{O}(d/\sqrt{T_{\text{ach}}})$.

Observe that for some $T_{\text{ach}} = \widetilde{\Theta}(d^2/((1 - 2\alpha)\epsilon - \Delta)^2)$, we have that $R_n \leq (1 - 2\alpha)\epsilon - \Delta$, where the conditions $\alpha < \frac{1}{2}$ and $\Delta < (1 - 2\alpha)\epsilon$ ensure that the bound on the regret is positive. This then implies that there exists a $t' \in \mathcal{T}$ such that

$$\max_{g \in \mathcal{G}_n} r^\star(g) - r^\star(g_{t'}) \leq (1 - 2\alpha)\epsilon - \Delta \tag{7}$$

Further, by Lemma C.7, there exists a $\tilde{g} \in \mathcal{G}$ such that $d^\star(g_{t'}, \tilde{g}) = \epsilon$ and

$$r^\star(\tilde{g}) - r^\star(g_{t'}) \geq (1 - 2\alpha)\epsilon. \tag{8}$$

Combining the above, we obtain

$$\begin{aligned}
r^\star(\tilde{g}) - \max_{g \in \mathcal{G}_n} r^\star(g) &\geq r^\star(\tilde{g}) - [(1 - 2\alpha)\epsilon - \Delta + r^\star(g_{t'})] \quad \text{(Equation (7))} \\
&= r^\star(\tilde{g}) - r^\star(g_{t'}) - (1 - 2\alpha)\epsilon + \Delta \\
&\geq (1 - 2\alpha)\epsilon - (1 - 2\alpha)\epsilon + \Delta \quad \text{(Equation (8))} \\
&= \Delta.
\end{aligned}$$

$\square$

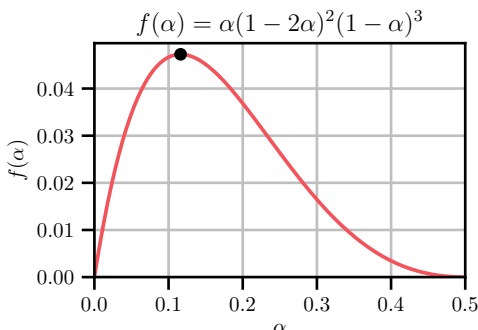

Figure 7: Plotting the effect of the parameter $\alpha$ on the (inverse) goal-achieval rate (cf. Theorem C.9). The target goal is reached fastest for $\alpha \approx 0.1$.

**Theorem C.9.** *Let Assumptions C.1 to C.5 hold with $\beta_t$ as in Equation (6). Fix any $\delta \in (0,1)$, $\alpha \in (0, \frac{1}{2})$, and let $D = d^\star(s_0, g^\star)$. Then, with probability $1 - \delta$, selecting goals $g_t$ with DISCOVER($\alpha, \beta_t$), the number of episodes $N$ until $g^\star \in \mathcal{G}_N$ is bounded by $N \leq \widetilde{O}(\frac{Dd^2}{\alpha(1-2\alpha)^2(1-\alpha)^3\kappa^3}) = \widetilde{O}(\frac{Dd^2}{\kappa^3})$.*

*Proof.* We first note that

$$r^\star(g^\star) - r^\star(s_0) = \alpha(V^\star(s_0, g^\star) - V^\star(s_0, s_0)) + (1-\alpha)(V^\star(g^\star, g^\star) - V^\star(s_0, g^\star))$$
$$= \alpha V^\star(s_0, g^\star) - (1-\alpha)V^\star(s_0, g^\star)$$
$$= (2\alpha - 1)V^\star(s_0, g^\star) = (1-2\alpha)d^\star(s_0, g^\star) = (1-2\alpha)D.$$

We prove the theorem by applying Lemma C.8 $M \stackrel{\text{def}}{=} \lceil \frac{(1-2\alpha)D}{\Delta} \rceil$ times, while setting $\epsilon = (1-\alpha)\kappa$. First, for an arbitrary $0 \leq i \leq M-1$, we assume for the goal set $\mathcal{G}_{iT}$ with some $T = \widetilde{\Theta}(\frac{d^2}{\alpha((1-2\alpha)(1-\alpha)\kappa - \Delta)^2})$ that it holds that

$$\max_{g \in \mathcal{G}_{iT}} r^\star(g) \geq r^\star(s_0) + i\Delta.$$

Now, applying Lemma C.8 yields that after an additional $T$ steps, with high probability, there exists a $t' \in \{iT, \ldots, (i+1)T\}$ such that there is a $\tilde{g} \in \mathcal{G}$ with $d^\star(g_{t'}, \tilde{g}) \leq (1-\alpha)\kappa$ satisfying

$$r^\star(\tilde{g}) - \max_{g \in \mathcal{G}_{iT}} r^\star(g) \geq \Delta.$$

Hence, by Assumption C.5, we have that $\tilde{g} \in \mathcal{G}_{(i+1)T}$, and therefore,

$$\max_{g \in \mathcal{G}_{(i+1)T}} r^\star(g) \geq r^\star(\tilde{g}) \geq \Delta + \max_{g \in \mathcal{G}_{iT}} r^\star(g) \geq r^\star(s_0) + (i+1)\Delta.$$

Iterating this argument $M$ times and applying a union bound, we obtain

$$\max_{g \in \mathcal{G}_{MT}} r^\star(g) \geq r^\star(s_0) + M\Delta \geq r^\star(g^\star).$$

The total number of episodes is

$$N \stackrel{\text{def}}{=} MT \leq \widetilde{O}\left( \frac{(1-2\alpha)Dd^2}{\Delta\alpha((1-2\alpha)(1-\alpha)\kappa - \Delta)^2} \right).$$

We can optimize $\alpha$ and $\Delta$ to minimize $N$ under the constraints $0 < \alpha < \frac{1}{2}$, $\Delta > 0$, and $\Delta < (1-2\alpha)(1-\alpha)\kappa$. The optimal choices are $\Delta = \frac{1}{3}(1-2\alpha)(1-\alpha)\kappa$ and $\alpha \approx 0.1$. Substituting, we obtain $N \leq \widetilde{O}(\frac{Dd^2}{\alpha(1-2\alpha)^2(1-\alpha)^3\kappa^3}) = \widetilde{O}(\frac{Dd^2}{\kappa^3})$. $\qquad \square$

Finally, we include a technical lemma that is used in the proof of Lemma C.8.

---

[6]See Lemma C.10.

**Lemma C.10.** *Let $0 < \alpha \leq 1$, $0 < \delta < 1$ and suppose $T_{\mathrm{ach}} \geq 8 \log(1/\delta)$. Furthermore, let $X_1, \ldots, X_T \overset{\mathrm{i.i.d.}}{\sim} \mathrm{Bern}(\alpha)$ and $S_T = \sum_{t=1}^T X_t$. Then, for some $T = \widetilde{\Theta}(\frac{T_{\mathrm{ach}}}{\alpha})$, with probability $1 - \delta$, we have $S_T \geq T_{\mathrm{ach}}$.*

*Proof.* Set

$$\gamma = \sqrt{\frac{2 \log(1/\delta)}{T_{\mathrm{ach}}}}, \quad T = \left\lceil \frac{(1+\gamma)^2\, T_{\mathrm{ach}}}{\alpha} \right\rceil = \widetilde{\Theta}\big(\tfrac{T_{\mathrm{ach}}}{\alpha}\big), \quad \mu = \mathbb{E}[S_T] = \alpha\, T.$$

Note that $\mu \geq (1+\gamma)^2 T_{\mathrm{ach}}$ and $0 < \gamma < 1$ (for $T_{\mathrm{ach}} \geq 8 \log(1/\delta)$). Hence,

$$T_{\mathrm{ach}} = (1 - \epsilon)\, \mu, \quad \epsilon = 1 - \tfrac{T_{\mathrm{ach}}}{\mu} = \tfrac{\gamma(2+\gamma)}{(1+\gamma)^2} \geq \tfrac{\gamma}{1+\gamma} \in (0, 1).$$

By the multiplicative Chernoff bound,

$$\Pr\big[S_T < T_{\mathrm{ach}}\big] = \Pr\big[S_T < (1-\epsilon)\mu\big] \leq \exp\big(-\tfrac{\epsilon^2 \mu}{2}\big) \leq \exp\big(-\tfrac{\gamma^2 T_{\mathrm{ach}}}{2}\big) = \exp\big(-\log(1/\delta)\big) = \delta.$$

$\square$

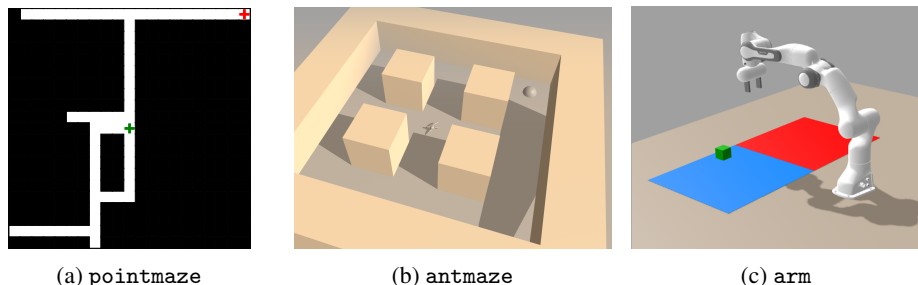

| (a) `pointmaze` | (b) `antmaze` | (c) `arm` |

Figure 8: Sparse-reward environments from the JaxGCRL [9] library used in our evaluation. We additionally implement the `pointmaze` environment (left), which allows for arbitrary dimensionality. The maze is created by randomly generating paths in the environment until the target is found sufficiently often.

# D   Additional Experimental Results

In this section, we present additional experiments and ablations. The environments used for this evaluation are visualized in Figure 8 [9].

**In high-dimensional search spaces, even direction estimates with high variance are useful.**   In complex environments, obtaining accurate direction estimates can be challenging. To evaluate the utility of directed goal selection in scenarios where direction estimates are imprecise, we add Gaussian noise to the target goal location. We then compare the number of environment steps required to reach a $10\%$ target goal achievement rate using the hand-designed goal selection strategy from Figure 5, under varying levels of noise variance. The results in Figure 9 show that even with substantial noise, `pointmaze` environments that are unsolvable by undirected methods, remain solvable by DISCOVER. This demonstrates that even imprecise directional estimates can significantly aid target goal discovery in complex goal spaces.

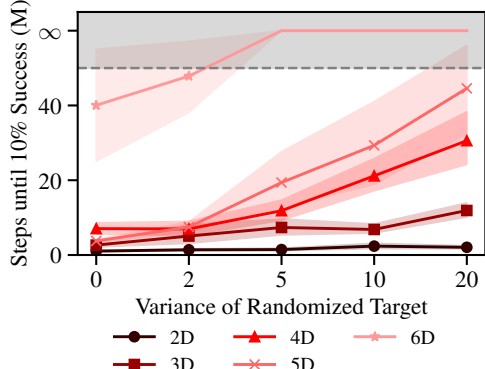

Figure 9: Evaluation of robustness directed goal selection with respect to noisy target estimates. The variance refers to the variance of the random Gaussian noise that is added to the target goal before selecting the goals using the hand-designed goal selection strategy, which uses the $L_2$ distance to estimate direction.

**Ablation of the online parameter adaptation strategy**   In Figure 10, we evaluate the effect of using the simple proposed adaptation strategy with different target goal achievement ratio $p_A^\star$, and compare with fixing the $\alpha_t$ parameters to 0 (Target Relevance + Novelty) and 0.5 (Fixed DISCOVER). Furthermore, we report the average $\alpha_t$ for DISCOVER for all easy and hard tasks respectively in Figure 11. The comparison of the success rates on the `antmaze` environments demonstrates that the adaptation strategy with any target goal achievement works better than fixing the parameters. This can be observed from the goal achievement rates. If we fix $\alpha_t = 0.5$, we choose goals that are "too easy" and therefore don't explore sufficiently. On the other hand, by fixing $\alpha_t = 0$ we select goals that are "too hard", which also leads too poor improvement. By using the simple adaptation strategy, we roughly achieve the target goal achievement specified. The optimal performance is achieved for the target goal achievement $p_A^\star = 0.5$, which is in line with what other methods found [53, 41, 84]. The average $\alpha_t$, which is found by the adaptation strategy, initially goes up to $0.15$, which is roughly what we found in the theoretical analysis in the linear bandit setting (cf. Theorem C.9), and then starts to decay. The decay can be explained by the fact that once we can reach the target goal, we don't need to optimize for achievability anymore.

**Influence of the term $\sigma(g, g^\star)$**   In Figure 12, we study how the standard deviation $\sigma(g, g^\star)$ from goal $g$ to the target $g^\star$ influences the training. This term theoretically is part of the UCB term, directing the agent towards the target goal. To this end, we fix the contribution of $\sigma(s_0, g)$ (i.e., set

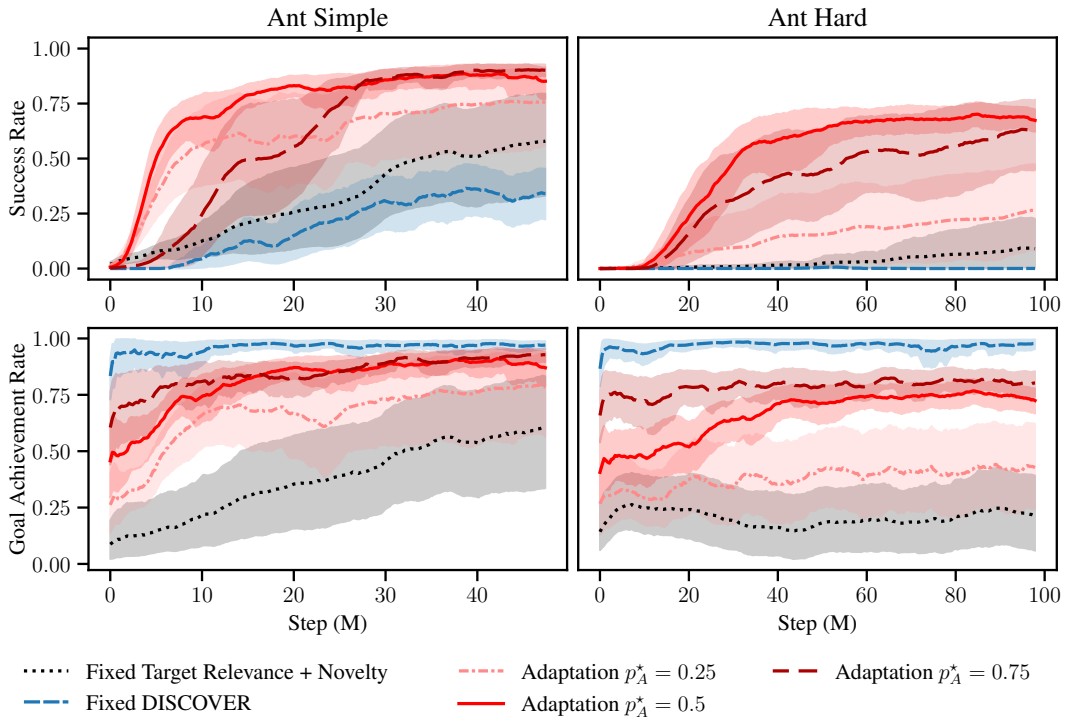

Figure 10: Comparison of how the adaptation strategy influences the goal achievement and success rates. We compare two constant strategies (Fixed DISCOVER and Fixed Target Relevance + Novelty) with an adaptation rule for different goal achievement targets.

d

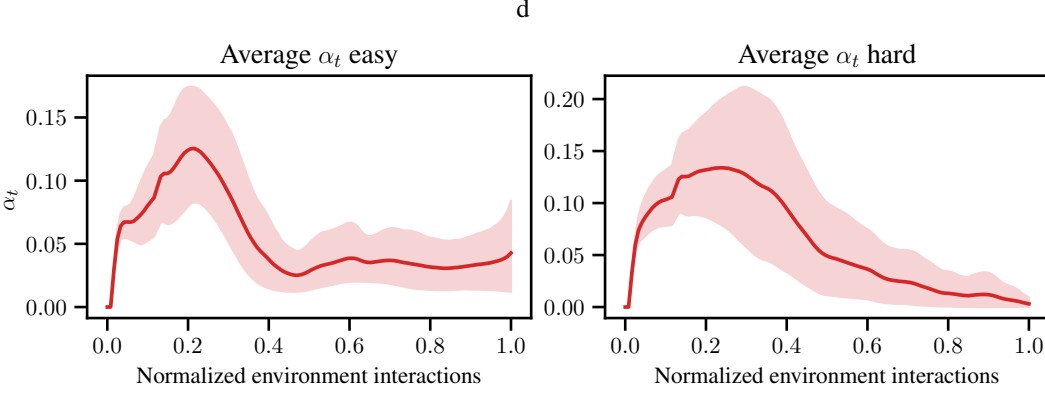

Figure 11: We plot the average $\alpha_t$ over the training, as adapted by the previously introduced online adaptation strategy for the DISCOVER goal selection strategy. The $\alpha_t$ are averaged over the three main environments.

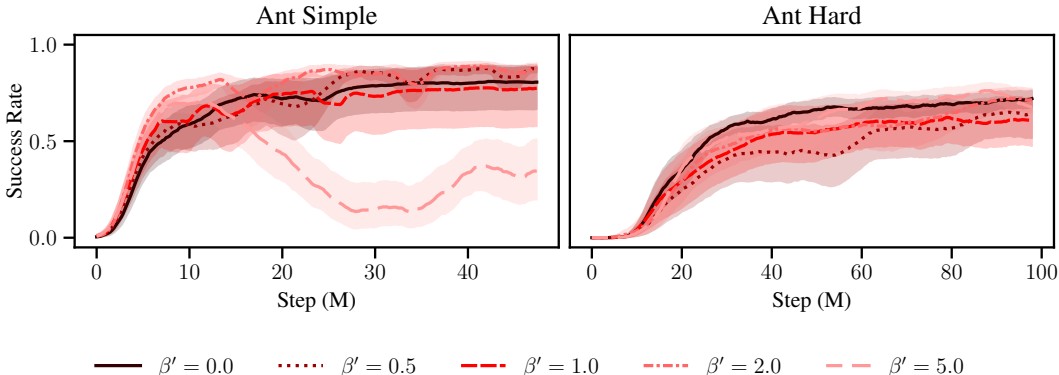

Figure 12: Comparison of how changing the coefficient of $\sigma(g, g^\star)$ influences the performance, when training on the two `antmaze` configurations. We use the configuration with $\beta' = 0$ in all other experiments.

$\beta_t = \frac{1}{\alpha_t}$) and consider a separate fixed $\beta_t' = \frac{\beta'}{1-\alpha}$, which determines the contribution of $\sigma(g, g^\star)$. The plot shows that no value for the $\beta'$ parameter has a significant positive effect on the performance. For this reason, we substitute $\sigma(s_0, g)$ for $\sigma(g, g^\star)$ in our other experiments with DISCOVER.

**Exploration of DISCOVER + pre-trained prior** We visualize the exploration of the DISCOVER + pre-trained prior gaol selection strategy in Figure 13. In comparison to DISCOVER starting from a randomly initialized agent, it only explores in the correct direction, avoiding obstacles. This demonstrates that access to prior can further improve performance of DISCOVER.

**Investigation of the role of the DISCOVER components for exploration** We visualize the different components of the DISCOVER objective over the course of training in Figure 14. The first term $V(s_0, g)$ has high-value close to the initial state. By maximizing it, we will pick a goal that is close to the start and likely to be *achievable*, which matches the intuition. The second term $V(g, g^\star)$ represents the value from a goal $g$ to the target goal $g^\star$.

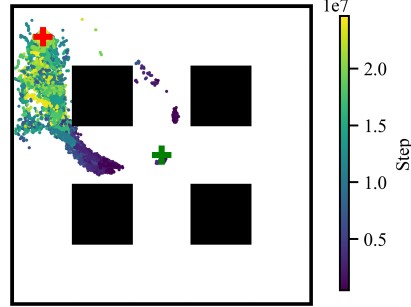

Figure 13: Selected goals by "DIS-COVER + pre-trained prior" in the `antmaze` environment.

The plots show that in the first episodes the value is small everywhere and only once goals are discovered that are closer to the final target we observe higher values. Over the course of training, its role of encouraging to pick goals *relevant* to the final target becomes more evident. This term therefore directs the goal selection towards the final goal. Finally, the standard deviation $\sigma(s_0, g)$ has the largest value at the border of the current achievable goal set and therefore encourages selecting *novel* goals. In general, the components of the DISCOVER objective during the training match the previously presented intuition and can efficiently guide the goal selection towards the desired target.

**Investigation of Direction Estimation** To further investigate the ability of the relevance term in the DISCOVER objective to direct the exploration towards to the target goal, we plot the pearson correlation between the relevance term $V^\star(g, g^\star)$ and a proxy for the true distance $||g - g^\star||$ in Figure 16a. The correlation in both the `antmaze` and `pointmaze` environments increases quickly during training. This indicates that the relevance term captures a useful notion of distance, even before the final goal is reached for the first time. This behavior explains DISCOVER's superior performance compared to undirected goal-selection strategies, as the value function is able to capture a useful notion of distance to the target, before it was reached. This allows DISCOVER to effectively guide the exploration to the true target goal. Notably, the correlation does not converge to one. This is expected, as the Euclidean distance in the goal space is an imperfect proxy for the true shortest traversable path within the environment.

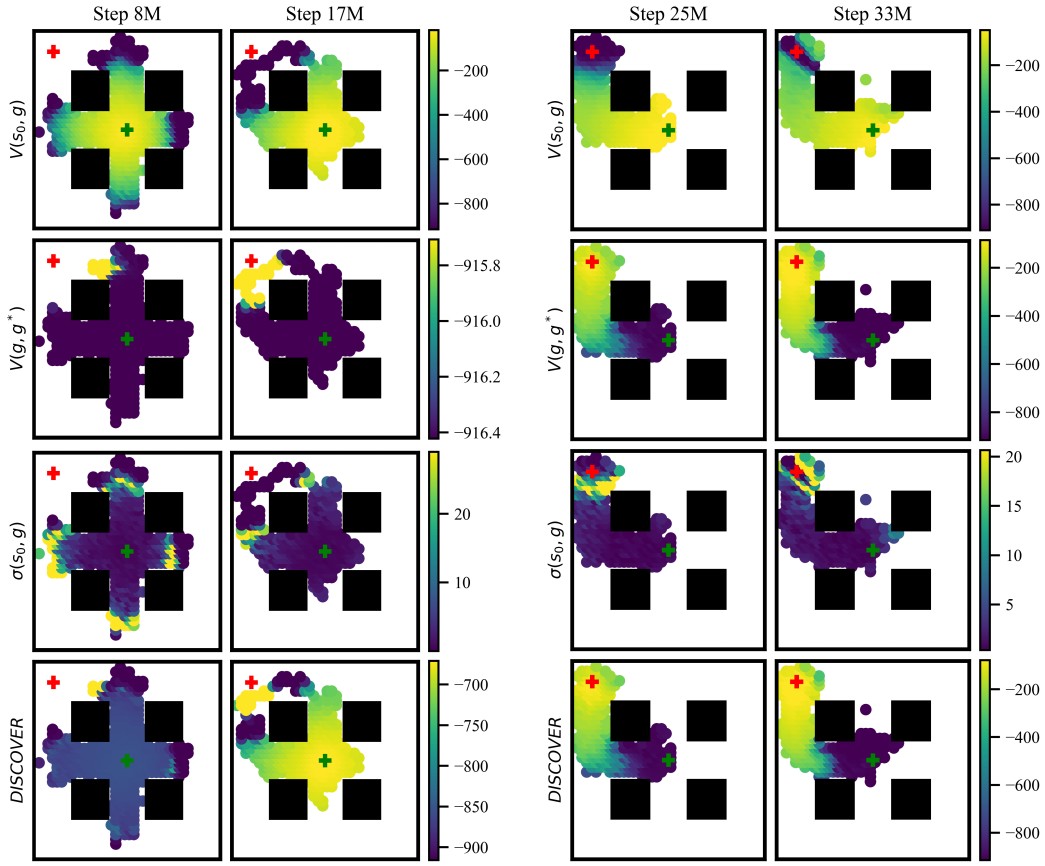

Figure 14: Visualization of the components of the DISCOVER objective at different points of the training. We plot the value functions in the regions, where achieved goals are sampled and therefore goals can be selected. The final DISCOVER objective combines the visualized terms with the current adaptation parameters $\alpha_t, \beta_t$.

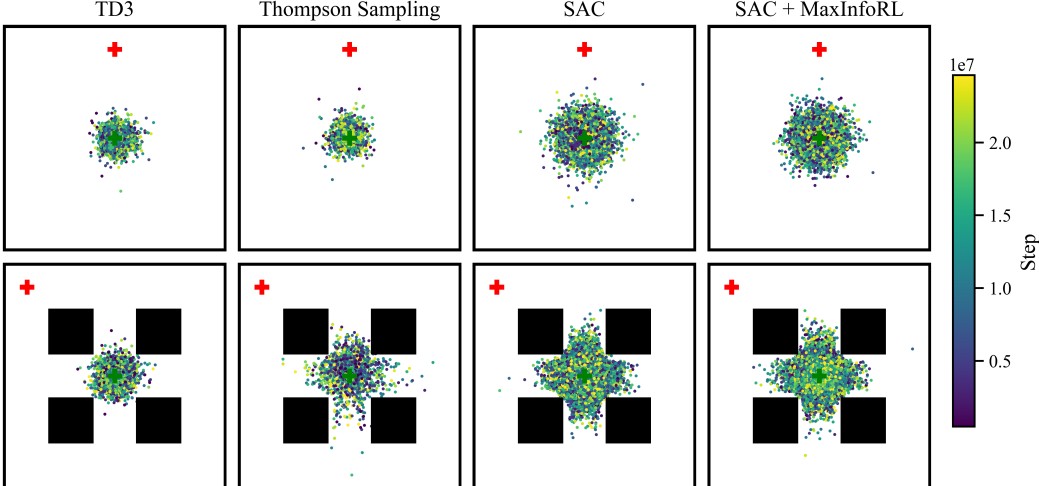

Figure 15: Visualization of the exploration of the standard non-goal-conditioned RL methods in the `antmaze` environments. We run the state-of-the-art MaxInfoRL [71] with SAC.

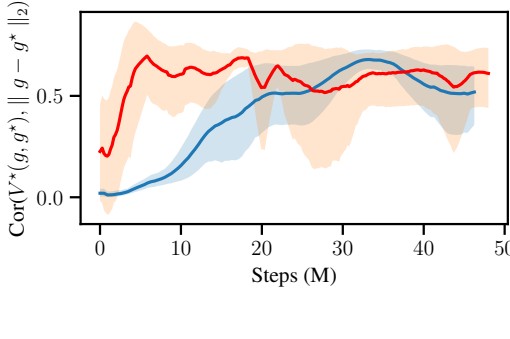 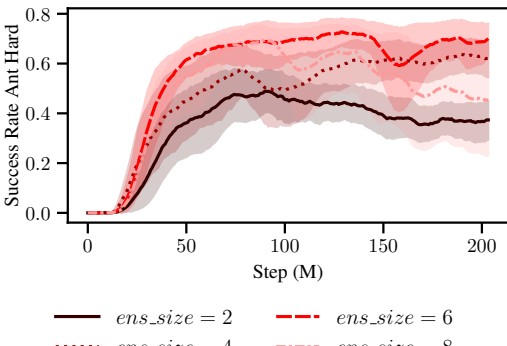

(a) Direction estimates become useful before the final goal is reached. We plot the pearson correlation between the current relevance estimate and the L2-distance between the goal and the target goal for the hard variants of the `antmaze` and `pointmaze` environments.

(b) A value function ensemble size of six strikes a good balance between computational cost and the quality of uncertainty estimates, leading to improved exploration and task performance. We plot the performance of DISCOVER varying the number of value function ensemble members in the hard `antmaze` environment.

**Ablation of Ensemble Size**   A key parameter in the DISCOVER algorithm is the value function ensemble size. To quantify the impact of ensemble size on uncertainty estimates, we evelute the performance of DISCOVER in the hard `antmaze` environment, comparing ensembles of 2, 4, 6, and 8 critics in Figure 16b. Consistent with prior work [48], increasing the number of critics above the default of two improves uncertainty estimation. In our experiments, an ensemble of six critics proved sufficient to yield robust performance.

Relying on an ensemble of critics can be computationally expensive. We find that in our experiments the overhead of increasing the critic size from 2 to 6 was modest. This is mainly due to small critic networks and highly parallel training using jax. In practice, we find that increasing the ensemble from 2 to 6 critics was sufficient to yield high-quality uncertainty estimates, while incurring only a 26% increase in runtime.

# E   Implementation Details

We consistently substitute $\sigma(g, g^\star)$ by $\sigma(s_0, g)$ in all our experiments. In our evaluated environments, the empirical performace of DISCOVER is largely irrespective of $\sigma(g, g^\star)$ (cf. Figure 12).

## E.1   Probabilistic Value Estimation

A crucial component of DISCOVER is a probabilistic model of the value function, which can enable uncertainty-aware strategies. Fortunately, there are many options for probabilistic models of the value function [39, 48, 47]. For simplicity, we employ an ensemble of value functions [48] and quantify uncertainty via disagreement. This strategy has been used before to select goals, which have high exploration potential and therefore provide novel experiences [86]. Intuitively, these ensemble provide valid uncertainty estimates, as we use different random initilizations for the networks and train on different data. If the networks have been trained on a certain training sample sufficiently often, the different ensemble members will converge to the same value, while if a sample hasn't been observed yet or only a few times the discrepancy will be higher. The mean and standard deviations used for the DISCOVER objective are computed as follows:

$$V(s,g) = \frac{1}{N}\sum_{i=1}^{N} V_i(s,g) \qquad\qquad \sigma^2(s,g) = \frac{1}{N}\sum_{i=1}^{N}(V_i(s,g) - V(s,g))^2 \tag{9}$$

This can be seen as a straightforward extension of the standard twin critic approach [24]. We find that a slightly higher numer of critic improves accuracy of uncertainty estimates. We further find that by training each critic against a random minimum of two target critics we obtain sufficient diversity for good uncertainty estimates as well as circumvent the maximization bias [81]. Additionally, we use a softplus activation at the output of each critic to limit the values to negative values.

Scaling critic ensembles to domains with large models (e.g., language) is challenging. An exciting direction for future work is to explore DISCOVER with other tools for uncertainty quantification, such as epistemic neural networks [50].

## E.2   Environment Details

We adapt the `antmaze` and `arm` environments from the JaxGCRL library [9]. In both cases, we fix the initial-state distribution to be a uniform pertubation of a fixed intial state, and we fix a single target goal location per environment (marked by the red cross in Figure 4). In addition to these two challenging high-dimensional benchmarks, we implement a `pointmaze` environment with potentially arbitrary dimensionality $d$. In our experiements we consider `pointmazes` with $d \in [2,\dots,6]$. We construct the `pointmazes` as follows. We sample random paths in the $d$-dimensional hypercube, by starting from the origin and randomly changing direction with a probability of $16.6\%$. We terminate this process once the target goal is observed sufficiently often. This number was set to 2 to 4 depending on the dimensionality of the maze.

In our experiments we evaluate both "simple" and "hard" versions of each environment. For the `pointmaze`, the simple configuration is realized on a two-dimensional grid while the hard configuration uses a four-dimensional hypercube. In the `antmaze`, the simple setup contains no walls between start and goal, whereas the hard version introduces a single wall that blocks the direct route to the target (visualized in Figure 4). Finally, in the `arm` environment the simple scenario places one small obstacle that the manipulator must skirt around, while the hard variant includes two larger obstacles that require more precise cube maneuvering to reach the goal.

In the `pointmaze`, the goal space coincides with the full state space, covering all $d$ dimensions (for the simple maze and for the hard maze). Consistent with JaxGCRL [9], in the `antmaze` the goals are defined by the agent's planar-coordinates, and in the `arm` environment by the 3-dimensional coordinates of the cube.

### E.3 Training Hyperparameters

| Hyperparameter | Value |
|---|---|
| Offline RL algorithm | TD3 |
| Ensemble size | 6 |
| Discount factor | 0.99 |
| Batch size | 256 |
| Learning rate | $3 \cdot 10^{-4}$ |
| Policy update delay | 2 |
| Target critic Polyak factor | 0.005 |
| Relabel strategy | Uniform future: $70\%$, original: $30\%$ |
| Target critic computation | Minimum of two random target critics |
| Size of critic ensemble | 6 |
| Initial apdation parameter $\alpha_0$ | 0 |
| Horizon | 100-250 |
| Parameter adaptation lookback $k_{\text{adapt}}$ | 64-128 |

Table 2: Hyperparameters for training in JaxGCRL Environments.

