# OpenReview forum: "DISCOVER: Automated Curricula for Sparse-Reward Reinforcement Learning"
_NeurIPS.cc/2025/Conference — NeurIPS 2025 poster_

### Official Review · Reviewer_9aeB · 2025-06-30

**Clarity:** 2
**Significance:** 3
**Originality:** 3
**Rating:** 5
**Confidence:** 3

**Summary:**

The paper introduces DISCOVER, a modification of goal-conditioned reinforcement learning for sparse goal-directed tasks that utilizes a set of intermediate goals to accelerate exploration. It combines estimates of achievability and novelty with estimates of the distance between the current and target goal to the goal selection process. The goal selection is then biased to select intermediate goals that are reliably achievable, novel, and are estimated to be close to the target goal. The paper offers a theoretical bound of the number of episodes needed to achieve the target task and shows the advantage of the proposed method in multiple experiments over three different environments.

**Questions:**

- What are the goals used in the experiments?

- How is the relevance term used before reaching the target goal for the first time?

- Why is the weighting term $\beta_t$ of the variance the same for both novelty and relevance? Also, is it set to $\beta_t=1$ in the experiments (as described in line 171)?

- If the theoretical bound is independent of the set of goals, how is it advantageous over standard RL?

**Ethical Concerns:**

["NO or VERY MINOR ethics concerns only"]

**Final Justification:**

The missing details of the experiment setup have been provided. The raised concerns regarding the benefit of the relevance term and theoretical results (and therefore the potential impact) have been lifted by the additional insights from the rebuttal.

Accordingly, I have increased my score.

**Limitations:**

yes

**Paper Formatting Concerns:**

- The focus on goal-directed rewards instead of general sparse-rewards could be clarified in the introduction and abstract.

- Fig. 1: Which environment was used?

- Fig. 4: Might be better represented by a table.

- Fig. 5: What is the meaning of the color scheme?

- Insight 5 / Fig. 7: [47] is cited but not part of the baselines

- Fig. 13 in Appendix: add dashes to better discern lines

- Table 1 in Appendix: discount factor repeated (as "discount factor" and "discounting factor")

**Quality:**

3

**Strengths And Weaknesses:**

### Quality
The method is well-motivated and validated in the experiments by comparing to three other baselines for goal selection, as well as to three baselines without goal-conditioning. The method is not compared to other methods that could use the goal information from hierarchical RL or reward shaping, but this comparison might be out-of-scope for this work.

A theoretical upper bound on the number of episodes learn to to reach the goal is given (informal intuition in the main text, formalized proofs in the appendix).

### Clarity
Overall, the paper is well-written. The motivation and description of the method are well-structured and comprehensible. The experiment setup lacks some details on the environments. Most notably the exact construction of the *pointmazes* environment (used dimensionalities and number of paths to goal) as well as the details on the *simple* and *hard* configurations for all environments including the used goals are missing. The result section is well structured and easy to follow.

### Significance
The method transfers general goal-conditioned RL to the sparse goal-directed setting to use intermediate goals to accelerate learning of a target task. The method is shown to improve over the state-of-the-art for goal selection strategies.

### Originality
To the best of my knowledge, the proposed method is novel and relevant citations are given where needed.

---

> ### Author Rebuttal · Authors · 2025-07-31
>
> We would like to thank you for a thorough review and for suggesting formatting and clarity improvements. We would like to address each point individually.
>
>
> ### Hierarchical RL and reward shaping
> > The method is not compared to other methods that could use the goal information from hierarchical RL or reward shaping, but this comparison might be out-of-scope for this work.
>
> We agree that goal-based settings are well suited for hierarchical methods or reward shaping. As pointed out, our work is focused specifically on the goal-selection problem driving exploration. As such, hierarchical and reward-shaping approaches are in part orthogonal, and can be combined to form even stronger methods. This remains an exciting avenue for future work.
>
> ### Environment Specification
> > The exact construction of the pointmazes environment (used dimensionalities and number of paths to goal)
>
> Thank you for this question! We consider 2 to 6 dimensional Point Mazes. We construct the Point Mazes as follows. We sample random paths in the $d$-dimensional hypercube, by starting from the origin and randomly changing direction with a probability of 16.6%. We terminate this process once the target goal is observed sufficiently often. This number was set to 2 to 4 depending on the dimensionality of the maze. We will add this clarification in Section E.2 of the Appendix.
>
> >  Details on the simple and hard configurations for all environments
>
> In our experiments we evaluate both “simple” and “hard” versions of each environment. For the Point Maze, the simple configuration is realized on a two-dimensional grid while the hard configuration uses a four-dimensional hypercube (as in Figure 4). In the Ant Maze, the simple setup contains no walls between start and goal, whereas the hard version introduces a single wall that blocks the direct route to the target. Finally, in the Arm environment the simple scenario places one small obstacle that the manipulator must skirt around, while the hard variant includes two larger obstacles that require more precise cube maneuvering to reach the goal. We will include all details in section E.2 of the appendix.
> > Fig. 1: Which environment was used?
>
> Thank you for pointing this out. Figure 1 reports the average success rate across all three tasks (Point Maze, Ant Maze, and Arm) separately for their simple and hard configurations. In other words, “Average Success Rate Simple/Hard” aggregates the mean performance shown in Figure 3 over those three environments. We will update the caption of Figure 1 to make this explicit.
>
> > What are the goals used in the experiments?
>
> In the Point Maze, the goal space coincides with the full state space, covering all d dimensions ($n=2$ for the simple maze and $d=4$ for the hard maze). Consistent with JAXGCRL [1], in the Ant Maze the goals are defined by the agent’s planar $(x,y)$-coordinates, and in the Arm environment by the $3$-dimensional coordinates of the cube.
> Thank you for highlighting these additional environment specifications, and will include all details in Section E.2 of the Appendix.
>
> ### Relevance term
> > How is the relevance term used before reaching the target goal for the first time?
>
> We appreciate this insightful question, which touches on a core aspect of DISCOVER. As you observed, in a strictly tabular setting, the relevance term would not be informative, as it would not be updated until the target goal is reached. However, in practice, the critic is a neural network and thus capable of some degree of generalization. We find that this generalization is not precise enough to extract an optimal policy for the target goal directly, but informative enough for subgoal selection. This knowledge is sufficient to rank subgoals by their expected utility toward the unseen target, effectively directing exploration.
>
> To substantiate this hypothesis, we have prepared an ablation that tracks the Pearson correlation between value estimates $V(\cdot, g^\star)$ and a reasonable proxy for the optimal value function (i.e., negative L2 distance between the relevant part of the state and the target goal). We find that even before the agent reaches the target once, the critic’s estimates increasingly align with this distance-based proxy. This shows that the value function can be used to determine which goals will be likely to lead to the target. For a visualization of this behavior we refer to Figure 15 in the appendix, showing that before the target goal is reached closer goals are determined to be of higher utility.
>
> | $Cor(V^\star(g,g^\star),\parallel g-g^\star\parallel_2)$ | 2M | 5M | 10M | 15M | 20M |
> |:------------------|-----------:|-----------:|-----------:|-----------:|------------:|
> | Ant Maze Hard | 1% |    4% | 13 % | 35 % | 48 % |
> | Point Maze Hard | 21%| 61 % | 60 % | 61 % | 61 % |
>
>
> > Why is the weighting term $\beta_t$  of the variance the same for both novelty and relevance? Also, is it set to $\beta_t = 1$ in the experiments?
>
> We fix $\beta_t =1$, which we find to ensure sufficient exploration in our experiments. We ablate the weighting of novelty and relevance in Figure 13 of the appendix (Section D). We find that choosing a fixed $\beta_t$ leads to similarly strong performance as adapting $\beta_t$ for each term. Intuitively, both uncertainty quantities are from the same critic; hence, one can expect them to be of similar scale.
>
> ### Theoretical bound
> > If the theoretical bound is independent of the set of goals, how is it advantageous over standard RL?
>
> Thank you for this question! We emphasize that the theoretical bound is independent of the goal set because we show that the final target becomes reachable in a number of steps proportional to the optimal distance to the target, yet independent of the goal space’s dimensionality. In contrast, standard RL with undirected exploration incurs complexity that scales with the volume of the goal set, making long-horizon tasks in high-dimensional spaces far more expensive [3]. We note that our theoretical results hold under the simplifying assumptions C.1–C.5.
> ### Formatting Concerns
>
> > Fig. 5: What is the meaning of the color scheme?
>
> The color scheme in Figure 5, as mentioned in the caption, visualizes the time at which a goal was selected. We will incorporate this information into the plot for clarification.
>
> > Insight 5 / Fig. 7: [47] is cited but not part of the baselines
>
> Thank you for pointing this out. [2] is closely related to the goal selection mechanism of Achievability + Novelty, as this strategy selects goals with high disagreement. We will adapt the citation accordingly.
>
> > The focus on goal-directed rewards instead of general sparse-rewards could be clarified in the introduction and abstract.
>
> > Fig. 4: Might be better represented by a table.
>
> > Fig. 13 in Appendix: add dashes to better discern lines
>
> > Table 1 in Appendix: discount factor repeated (as "discount factor" and "discounting factor")
>
> Thank you for reviewing our submission thoroughly and your suggestions! We are happy to directly fix all of these issues, namely by making the reward class explicit in the introduction, replacing Fig. 4 with a table, and updating the Appendix as suggested.
>
> —
>
> We hope we were able to clarify your concerns, and look forward to the coming discussion window. Please let us know if you have any further questions or suggestions. We would greatly appreciate it if you could reconsider the score based on our response and extended results.
>
>
> ### References
> [1] Bortkiewicz et al., Accelerating Goal-conditioned Reinforcement Learning Algorithms and Research, ICLR 2025
>
> [2] Ian Osband, Charles Blundell, Alexander Pritzel, and Benjamin Van Roy. Deep exploration via
> bootstrapped DQN. In NeurIPS, 2016
>
> [3]  Jean Tarbouriech, Omar Darwiche Domingues, Pierre Menard, Matteo Pirotta, Michal Valko,
> and Alessandro Lazaric. Adaptive multi-goal exploration. In AISTATS, 2022

---

> > ### Comment · Reviewer_9aeB · 2025-08-05
> >
> > Thank you for the insightful explanations! My concerns regarding the benefit of the relevance term and the theoretical results have been lifted. Accordingly, I will raise my score.

---

> > > ### Author Response · Authors · 2025-08-05
> > >
> > > Thank you for your thorough, actionable feedback and your positive evaluation! We are glad the rebuttal was instrumental in highlighting the significance of the relevance component, and the theoretical results.

---

### Official Review · Reviewer_nh3j · 2025-07-03

**Clarity:** 4
**Significance:** 4
**Originality:** 4
**Rating:** 6
**Confidence:** 4

**Summary:**

This paper presents DISCOVER, a goal-directed exploration method for RL, which performs exploration using a combined metric that defines a set of goals to attempt so as to focus exploration towards the true target goal state.

**Questions:**

1. The discussion of the value of "sense of direction" in the abstract and figure 1 is confusing, since this isn't standard exploration terminology and thus without a definition on hand leaves the reader confused as to what is meant. Consider writing those passages.

2. Does the use of the value estimate for the achievability component of the DISCOVER objective result in favoring goals reachable in fewer timesteps? If not, what prevents this bias?

3, In section 4.2, how do these value estimates affect exploration? Maximizing value is simple exploitation.

4. In theorem 4.2 and associated sections, I'm confused as to why the bound is linear with the distance to the target. The value function has no knowledge of the target, so why would it bias selected goals towards the target? The target has never been observed during training, by construction, no? I must be missing something about how there is useful information about an unknown target in the value function. Is this relying on generalization in function approximation? I'm unsure why that would appear in the theoretical bound, though.

5. In figure 7, it seems odd to compare primarily to TD3 and SAC, since these are not really exploration methods. Ideally I'd suggest comparing to random network distillation here, as a commonly used baseline for a state novelty based exploration bonus method.

**Ethical Concerns:**

["NO or VERY MINOR ethics concerns only"]

**Final Justification:**

The authors were able to provide convincing intuition for why the proposed method works so well, as well as adding an important comparison method (which performed poorly compared to DISCOVER). As such I raised my score since I have no remaining concerns and I think this is potentially high-impact work that can break new ground on exploration in RL, a critically important challenge in the field for which few effective and general methods exist presently.

**Limitations:**

Limitations are suitably discussed. I don't see any direct potential negative societal impacts from this work.

**Paper Formatting Concerns:**

I don't see any major paper formatting issues.

**Quality:**

3

**Strengths And Weaknesses:**

Overall, I think this paper is excellent. It is very well written (easily the best read in my batch of papers to review for NeurIPS this cycle), strongly motivated by an important topic in RL, and shows strong performance from an intuitive and theoretically grounded method for high impact. While similar in some ways to prior goal directed exploration methods, the key "relevance" term proposed is to my knowledge novel, and the paper does a good job of grounding the other elements of the algorithm in prior work.

The main weakness I can see is that I'm still a bit confused as to why this method works so well- Given that the true goal is never sampled during early training for the evaluated tasks, I'm a little confused as to how the relevance term is providing useful information for goal selection. Some of this might be down to imperfect explanation, but I can imagine experiments that would shine light on when and for what states or goals this term provides a meaningful signal and when/where it is basically noise (perhaps comparing the relevance term computed with the current learned value function to a known oracle/pretrained value function for different states/goals?).

In aggregate, though, I think this paper is a quality submission and more than sufficient for publication as is. If my concern about the basis for outperformance in the experimental results can be addressed and added to the paper, I would raise my score even further as well.

---

> ### Author Rebuttal · Authors · 2025-07-31
>
> We would like to thank you for the positive evaluation of our submission, and the detailed and clear feedback. We will address each point individually.
>
> > Overall, I think this paper is excellent.
>
> Thank you for this evaluation!
>
> ### Relevance and sense of direction
>
> > I'm a little confused as to how the relevance term is providing useful information for goal selection
>
> We appreciate this insightful question, which touches on a core aspect of DISCOVER. As you observed, in a strictly tabular setting the relevance term would not be informative, as it would not be updated until the target goal is reached. However, in practice, the critic is a neural network and thus capable of some degree of generalization. We find that this generalization is not precise enough to extract an optimal policy for the target goal directly, but informative enough for subgoal selection. This knowledge is sufficient to rank subgoals by their expected utility toward the unseen target, effectively directing exploration.
>
> To substantiate this hypothesis, we have prepared an ablation that tracks the Pearson correlation between value estimates $V(\cdot, g^\star)$ and a reasonable proxy for the optimal value function (i.e., negative L2 distance between the relevant part of the state and the target goal). We find that even before the agent reaches the target once, the critic’s estimates increasingly align with this distance-based proxy. This shows that the value function can be used to determine which goals will be likely to lead to the target. For a visualization of this behavior we refer to Figure 15 in the appendix, showing that before the target goal is reached closer goals are determined to be of higher utility.
>
> | $Cor(V^\star(g,g^\star),\parallel g-g^\star\parallel_2)$ | 2M | 5M | 10M | 15M | 20M |
> |:------------------|-----------:|-----------:|-----------:|-----------:|------------:|
> | Ant Maze Hard | 1% |    4% | 13 % | 35 % | 48 % |
> | Point Maze Hard | 21%| 61 % | 60 % | 61 % | 61 % |
>
>
> > The discussion of the value of "sense of direction" in the abstract and figure 1 is confusing
>
> Thank you for highlighting this! We agree and will include the following sentences in the abstract to clarify the meaning of “sense of direction”:
>
> We argue that solving such challenging tasks requires solving simpler tasks that are relevant to the target task, i.e., whose achievement will teach the agent skills required for solving the target task.
> We demonstrate that this sense of direction, necessary for effective exploration, can be extracted from existing RL algorithms, without leveraging any prior information.
>
> ### Achievability component
>
> > Does the use of the value estimate for the achievability component of the DISCOVER objective result in favoring goals reachable in fewer timesteps?
>
> Indeed, if the achievability component *alone* is considered, DISCOVER would select goals that can easily be reached. The two additional components (relevance and novelty, see Equation 3) are thus crucial to recover guarantees (Theorem C.9) and good empirical performance. In combination with them, this term mostly prevents “reckless” exploration, which would only pursue subgoals that are promising, but also very hard to reach.
>
>
> ### Exploration vs exploitation
>
> > In section 4.2, how do these value estimates affect exploration? Maximizing value is simple exploitation.
>
> Theorem 4.2 follows from our use of a probabilistic model over value estimates combined with an optimistic goal‐selection strategy (see lines 152–154). By acting *on the upper confidence bounds* of these estimates, the algorithm naturally incorporates the novelty term from Equation 3. In contrast, greedily maximizing value alone would amount to pure exploitation and yield suboptimal exploration. The upper confidence bound maximizes the mean as well as the standard deviation of the value functions, leading to the desired exploration.
>
> ### Linear bound
>
> > In theorem 4.2 and associated sections, I'm confused as to why the bound is linear with the distance to the target.
>
> Theorem 4.2 holds in a simplified setting, that is defined through Assumptions C.1 to C.5. In particular, the assumptions C.1 (linearity of the optimal value function) and C.2 (noisy feedback of the value between subgoal and target goal) enable the agent to learn the value function. Carefully balancing exploration and exploitation (via the UCB objective) then allows the agent to explore in only roughly the direction of the target goal. We recognize that assumptions C.1 and C.2 are simplifying and do not hold in most practical settings. The assumptions can be seen as a parallel to the generalization ability of a neural critic, as demonstrated in the correlation results above.
>
>
> ### RND baseline
>
> > I'd suggest comparing to random network distillation
>
> Thank you for suggesting this comparison, which we believe highlights the value of direction on sparse-reward tasks. We agree that intrinsic‐reward methods play an important role in sparse‐reward, long‐horizon tasks. By design, RND is task-agnostic, and will provide a strong signal towards sources of novelty. RND relies on an intrinsic reward term: within the framework of goal selection for exploration, MEGA (see Figure 2) can be seen as its counterpart, since it largely samples novel goals in any direction.
>
> To address your comment, we have evaluated RND as a baseline. We have evaluated RND on simple configurations of the three considered environments (Point Maze, Ant Maze and Arm) and **find that it achieves 0% success rate across all tasks**, while DISCOVER achieves on average 60%. This is in line with the results we report for the MaxInfoRL baseline, which is another intrinsic-reward method, based on information gain of a dynamics model. We will report the RND baselines, along the MaxInfoRL baseline, in Figure 7. These results underscore the advantage of our directed, goal-conditioned exploration approach, which is required to solve hard exploration tasks.
>
> —
>
> We would like to thank you for your thorough and positive evaluation. We hope to have adeptly answered your remaining questions. We hope that the additional experimental results were able to address the improved performance of our method sufficiently to justify increasing the score. Please let us know if you have any further questions or suggestions.
>
> [1] Silviu Pitis, Harris Chan, Stephen Zhao, Bradly Stadie, and Jimmy Ba. Maximum entropy gain
> exploration for long horizon multi-goal reinforcement learning. In ICML, 2020

---

> > ### Comment · Reviewer_nh3j · 2025-08-04
> >
> > Thanks for the improved intuition! This answer makes sense to me, and assuming it's clarified a little in the paper (shouldn't be a major revision), combined with the RND baseline I have no significant issues with this work.
> >
> > As such I have raised my score- this is potentially groundbreaking for exploration in RL, and I look forward to seeing future work build upon it in various directions.

---

> > > ### Author Response · Authors · 2025-08-05
> > >
> > > Thank you for your positive evaluation! We are also excited about pursuing future work in this direction and seeing others building on top of this work.

---

### Official Review · Reviewer_2Mok · 2025-07-03

**Clarity:** 3
**Significance:** 3
**Originality:** 3
**Rating:** 4
**Confidence:** 2

**Summary:**

This paper introduces DISCOVER, a novel goal-selection strategy for sparse-reward reinforcement learning that creates an automated curriculum of exploratory tasks. The method balances the achievability, novelty, and relevance of potential goals, using bootstrapped value function estimates to direct exploration towards a final target task without requiring prior information. The authors demonstrate that this approach significantly outperforms state-of-the-art exploration methods on 3 complex, long-horizon control tasks.

**Questions:**

If the bootstrapped estimate of relevance is not optimal, could it trap the agent into refining its policy within a suboptimal point, thereby preventing the discovery of a better path to the goal?

**Ethical Concerns:**

["NO or VERY MINOR ethics concerns only"]

**Final Justification:**

The authors have addressed most of my concerns during the rebuttal period.

**Limitations:**

yes

**Quality:**

3

**Strengths And Weaknesses:**

Strengths

- The problem of generating a curriculum of simpler, relevant tasks is crucial for solving complex, high-dimensional problems, and this paper makes a direct and effective contribution to this area.
- Bootstrapping a "sense of direction" entirely from the agent's own value estimates is a clever and practical approach. This avoids the need for strong prior knowledge or costly auxiliary components like generative models, which are often required by other directed exploration strategies.

Weakness

- Although the method is presented as general, the empirical evaluation is confined to navigation-centric environments (Point Maze, Ant Maze, Arm)  where a geometric "sense of direction" is intuitive. This casts some doubt on the method's generality, as the vital "relevance" component may not function as effectively in more abstract or combinatorial tasks that lack a simple spatial correlation between intermediate and final goals.

---

> ### Author Rebuttal · Authors · 2025-07-31
>
> Thank you for your review, and the positive evaluation of our work. We are happy to address each point.
>
> > Bootstrapping a "sense of direction" entirely from the agent's own value estimates is a clever and practical approach.
>
> Thank you!
>
> ### Generality of the DISCOVER goal selection method.
>
> > Although the method is presented as general, the empirical evaluation is confined to navigation-centric environments (Point Maze, Ant Maze, Arm) where a geometric "sense of direction" is intuitive.
>
> We appreciate this observation. We agree that, at a high level, the geometry of the tasks considered is rather intuitive. However, we would like to stress that the networks (and in particular the critic) largely receive proprioceptive measurements: in antmaze, only 2 of the 29 state dimensions convey information about the ant’s position in a 2D plane, while all remaining variables describe the configuration of the ant’s joints. Therefore, these representations are not overly biased towards a simple 2D geometry, and **the critic needs to learn the sense of direction from scratch**. Even when this information is isolated, the critic cannot rely on simple L2 distances, but needs to learn about the obstacle we introduced in the hard variants.
>
> Because our approach operates on learned representations rather than handcrafted objectives, we believe it holds the potential of  generalizing beyond spatial tasks to tasks with a less straightforward sense of direction. For instance, one could imagine applying the same directional exploration principles for reasoning tasks in large language models. Exploring such extensions is an exciting avenue for future work.
>
> ### Escaping local optima.
>
> > If the bootstrapped estimate of relevance is not optimal, could it trap the agent into refining its policy within a suboptimal point, thereby preventing the discovery of a better path to the goal?
>
> It is indeed true that relying exclusively on the mean value estimate could lead the agent to commit to a suboptimal direction. To address this, we adopt an Upper Confidence Bound (UCB) inspired selection criterion that explicitly encourages exploration by maximizing standard deviation of the value function ensemble. This leads to the method prioritizing novel experiences effectively and avoids getting stuck in local optima. Intuitively, after exploring a currently believed optimal path sufficiently often the value function will become more concentrated on the mean and therefore reduce its standard deviation. Hence the method will switch to alternative goals, which maximize the standard deviation leading to novel experiences. We empirically and theoretically show that this trade-off between exploration and exploitation leads to strong performance (Theorem 4.2, Figure 3). Balancing exploration and exploitation is crucial for preventing the agent from being trapped at a suboptimal point.
>
> —
>
> We thank you for raising these important points about our algorithm’s generality and exploration capabilities. We hope to have addressed your remaining concerns. Please let us know if you have any further questions or suggestions. We would greatly appreciate it if you could reconsider the score based on our response and extended results.

---

> > ### Comment · Reviewer_2Mok · 2025-08-06
> >
> > Thanks for your reply. Most of my concerns have been addressed, and I will maintain my assessment.

---

> ### Author Response · Authors · 2025-08-06
>
> Dear Reviewer 2Mok,
>
> We greatly appreciate your effort in reviewing our paper. We would like to double-check if our response was able to address each of the points you raised. Please let us know if anything remains unclear, we are happy to further discuss!

---

### Official Review · Reviewer_fisY · 2025-07-07

**Clarity:** 2
**Significance:** 2
**Originality:** 2
**Rating:** 5
**Confidence:** 2

**Summary:**

The paper introduces a method called "DISCOVER" that selects exploratory goals in the direction of task. Using the critic networks in deep learning, the method can propose achievable goals that can help an agent with a better goal selection to complete the task. By learning an ensemble of critics, the method can also estimate uncertainty regarding a task and leverages value estimates to find useful intermediate goals. The method shows considers three sparse-reward long-horizon settings (Point Maze, Ant Maze and Arm) and shows improvements over the baselines.

**Questions:**

- Can you please explain why the success rate is still low for almost half of the environment settings? Also, how exactly is the success rate calculated here?
- Have you considered comparing your method with the intrinsic reward based exploration methods that can do well over long-horizon settings?
- How sensitive are the uncertainty estimates based on the size of the ensembles used?

**Ethical Concerns:**

["NO or VERY MINOR ethics concerns only"]

**Final Justification:**

Update:
I would like to thank the authors for their rebuttal which has been useful for me to understand their work better. I have also looked at the reviews and discussions from other reviewers that has also clarified a bunch of my concerns. Since the authors have clarified most of my concerns, I am updating my scores accordingly.

**Limitations:**

Yes

**Quality:**

2

**Strengths And Weaknesses:**

Strengths:
- The method addresses an important problem of exploration over long horizons in RL and considers goal based RL settings.
- Using a combination of metrics (achievability, novelty and relevance) sounds a reasonable and interesting metric to consider goal selection.

Weaknesses:
- The method relies on an ensemble of critics which can be computationally expensive and slow to learn in challenging environments.
- The tasks used for evaluation are simpler tasks, as compared to more difficult to explore settings in which goal-based methods can actually be helpful in achieving good exploration.
- The baselines considered do not consider the usual exploration based methods, especially those that are based on intrinsic rewards or RND rewards in sparse-reward settings. A comparison with those can help one understand if the goal-based methodology is even required given when these exploration based methods fail.
- The success rate of some of the tasks is still low, eg in Easy setting of Arm, and Hard setting of PointMaze and Arm. It would be interesting to see if the curves changes as more number of timesteps are added.

---

> ### Author Rebuttal · Authors · 2025-07-31
>
> Thank you for your review and detailed comments! We are happy to address each of the points individually.
>
> > Using a combination of metrics (achievability, novelty and relevance) sounds like a reasonable and interesting metric to consider for goal selection.
>
> Thank you!
>
> ### Computational cost of ensembles
>
> > The method relies on an ensemble of critics which can be computationally expensive and slow to learn in challenging environments
> We agree that large neural‐network ensembles can incur significant costs, particularly at the scale of today’s language models, but in our experiments the overhead was modest for three main reasons. First, training of the ensembles can be parallelized in modern machine learning frameworks such as JAX, substantially reducing wall-clock time. Second, deep RL methods typically employ relatively small critic networks (or even partial ensembles [1]), keeping the overall memory footprint low. Third, TD3 and the majority of off-policy RL algorithms already rely on twin critics, in order to control the overestimation bias. These three factors have allowed previous works to dramatically scale the ensemble size (up to 100 [2]).
> In practice, we find that  increasing the ensemble from 2 to 6 critics was sufficient to yield high‐quality uncertainty estimates, while incurring only a ~26 % increase in runtime:
>
> |                   | ens_size=2 | ens_size=4 | ens_size=6 | ens_size=8 |
> |:------------------|-----------:|-----------:|-----------:|-----------:|
> | **Time for 1M steps** |     109.8s |     121.5s |     138.6s |     144.6s |
>
>
> > How sensitive are the uncertainty estimates based on the size of the ensembles used?
>
> Thank you for suggesting this important ablation! To quantify the impact of ensemble size on uncertainty estimates, we conducted an ablation study in the hard Ant Maze environment, comparing ensembles of 2, 4, 6, and 8 critics. Consistent with prior work [2], increasing the number of critics above the default of two improves uncertainty estimation. In our experiments, an ensemble of six critics proved sufficient to yield robust performance:
>
>
> | Ensemble Size                 | Success Rate @15M | @25M | @50M | @100M | @150M |
> |:------------------|-----------:|-----------:|-----------:|-----------:|------------:|
> | ens_size=2 | 0% |    7% | 36 % | 46 % | 40 % |
> | ens_size=4 | 2%| 19 % | 45 % | 50 % | 61 % |
> | ens_size=6 | 1%| 17 % | 61 % | 69 % | 66 % |
> | ens_size=8 | 1%| 14 % | 54 % | 63 % | 55 % |
>
>
>
> These results demonstrate that using six critics strikes a good balance between computational cost and the quality of uncertainty estimates, leading to improved exploration and task performance.
>
> ### Comparison to RND rewards
>
> > The baselines considered do not consider the usual exploration based methods, especially those that are based on intrinsic rewards or RND rewards
>
> > Have you considered comparing your method with the intrinsic reward based exploration methods that can do well over long-horizon settings?
>
> Thank you for suggesting this comparison, which we believe highlights the value of direction on sparse-reward tasks. We agree that intrinsic‐reward methods play an important role in sparse‐reward, long‐horizon tasks. By design, RND is task-agnostic, and will provide a strong signal towards sources of novelty. RND relies on an intrinsic reward term: within the framework of goal selection for exploration, MEGA (see Figure 2) can be seen as its counterpart, since it largely samples novel goals in any direction.
>
> To address your comment, we have evaluated RND as a baseline. We have evaluated RND on simple configurations of the three considered environments (Point Maze, Ant Maze and Arm) and **find that it achieves 0% success rate across all tasks**, while DISCOVER achieves 60% on average. This is in line with the results we report for the MaxInfoRL baseline, which is another intrinsic-reward method, based on information gain of a dynamics mode. Both RND and MaxInfoRL explore into “all” directions until a reward is observed for the first time, which leads to inefficient exploration. We will report the RND baselines, along the MaxInfoRL baseline, in Figure 7. These results underscore the advantage of our directed, goal-conditioned exploration approach, which is required to solve hard exploration tasks.
>
> ### Complexity of tasks and asymptotic success rate
>
> > Can you please explain why the success rate is still low for almost half of the environment settings?
>
> The set of environments we consider was recently introduced as part of JAXGCRL [3], and was thus designed to be challenging for modern GCRL algorithms. Figure 3 from [3] shows that, except for the simplest environment (Reacher, which we do not consider), the remaining tasks are only partially solved given a limited compute budget. As our setup (single‐goal, sparse‐reward RL) differs from the multi‐goal experiments in JAXGCRL, this does not constitute an apple-to-apple comparison, but rather an indication of the challenging nature of the environments.
>
> > Also, how exactly is the success rate calculated here?
>
> The policy is periodically evaluated every 250-500K episodes, depending on the environment. Each episode is considered successful if any state within a finite horizon (100-250 depending on the difficulty of the task) is close enough to the goal, where the closeness criterion is the standard one defined by JAXGCRL [3] (L2-distance on the relevant part of the state).
>
> —
>
> We thank you for raising these important points, which we think highlight two important features of the algorithm, namely its computational footprint, and its performance gains over intrinsic reward approaches. We hope to have addressed your remaining concerns. Please let us know if you have any further questions or suggestions.  We would greatly appreciate it if you could reconsider the score based on our response and extended results.
>
>
> #### References
>
> [1] Sekar et al., Planning to Explore via Self-Supervised World Models, ICML 2020
>
> [2] An et al., Uncertainty-Based Offline Reinforcement Learning with Diversified Q-Ensemble, NeurIPS 2021
>
> [3] Bortkiewicz et al., Accelerating Goal-conditioned Reinforcement Learning Algorithms and Research, ICLR 2025

---

> ### Author Response · Authors · 2025-08-05
>
> Dear Reviewer fisY,
>
> We greatly appreciate your time and effort in reviewing our paper. We would like to double-check whether our response is adeptly addressing your concerns and questions. Please let us know if anything remains unclear, we are happy to further discuss!

---

### Decision · Program_Chairs · 2025-09-17

**Decision:**

Accept (poster)

**Comment:**

This paper introduces DISCOVER, a goal-selection strategy for sparse-reward RL that combines achievability, novelty, and relevance. By leveraging ensembles of critics to estimate uncertainty, the method builds automated curricula that guide exploration toward intermediate goals. Theoretical analysis provides connections to UCB-style exploration, and experiments on PointMaze, AntMaze, and Arm tasks show consistent improvements over existing goal-selection and standard RL baselines.

**Strengths:** Clear motivation, solid theoretical grounding, and good empirical results across multiple tasks. The idea of incorporating a relevance signal into goal selection is novel and practically effective. The paper is also clearly written and well structured.

**Weaknesses:** Evaluation is restricted to navigation-style tasks, leaving some doubt about generality. Comparisons with some intrinsic-reward exploration methods are also missing. The authors are encouraged to enhance their manuscript with at least some discussion on these issues.

**Final Recommendation:** Overall, DISCOVER represents a meaningful step forward in exploration for sparse-reward RL. Despite some limitations, the contributions are clear, the results compelling, and reviewer concerns were largely addressed in rebuttal. I recommend acceptance.